# Sparkles: Unlocking Chats Across Multiple Images for Multimodal Instruction-Following Models

## Abstract

Large language models exhibit enhanced zero-shot performance on various tasks when fine-tuned with instruction-following data. Multimodal instruction-following models extend these capabilities by integrating both text and images. However, existing models such as MiniGPT-4 face challenges in maintaining dialogue coherence in scenarios involving multiple images. A primary reason is the lack of a specialized dataset for this critical application. To bridge these gaps, we present **SparklesChat**, a multimodal instruction-following model for open-ended dialogues across multiple images. To support the training, we introduce **SparklesDialogue**, the first machine-generated dialogue dataset tailored for word-level interleaved multi-image and text interactions. Furthermore, we construct **SparklesEval**, a GPT-assisted benchmark for quantitatively assessing a model's conversational competence across multiple images and dialogue turns. Our experiments validate the effectiveness of SparklesChat in understanding and reasoning across multiple images and dialogue turns. Specifically, SparklesChat outperformed MiniGPT-4 on established vision-language benchmarks, including the BISON binary image selection task and the NLVR2 visual reasoning task. Moreover, SparklesChat scored 8.56 out of 10 on SparklesEval, substantially exceeding MiniGPT-4's score of 3.91 and nearing GPT-4's score of 9.26. Qualitative evaluations further demonstrate SparklesChat's generality in handling real-world applications. We have uploaded the code, model, and data as supplementary material to ensure reproducibility.

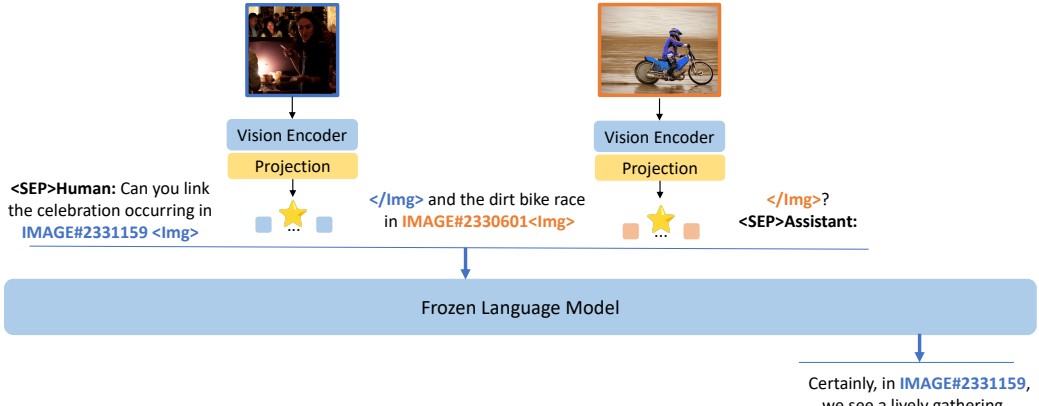

Figure 1: The architecture of SparklesChat. SparklesChat integrates multiple images at the word level within the dialogue, facilitating a fine-grained and human-like multimodal interaction.

# 1 Introduction

Large language models (LLMs) have shown remarkable progress in zero-shot performance across a variety of tasks when fine-tuned using instruction-following data (Ouyang et al., 2022; Openai,

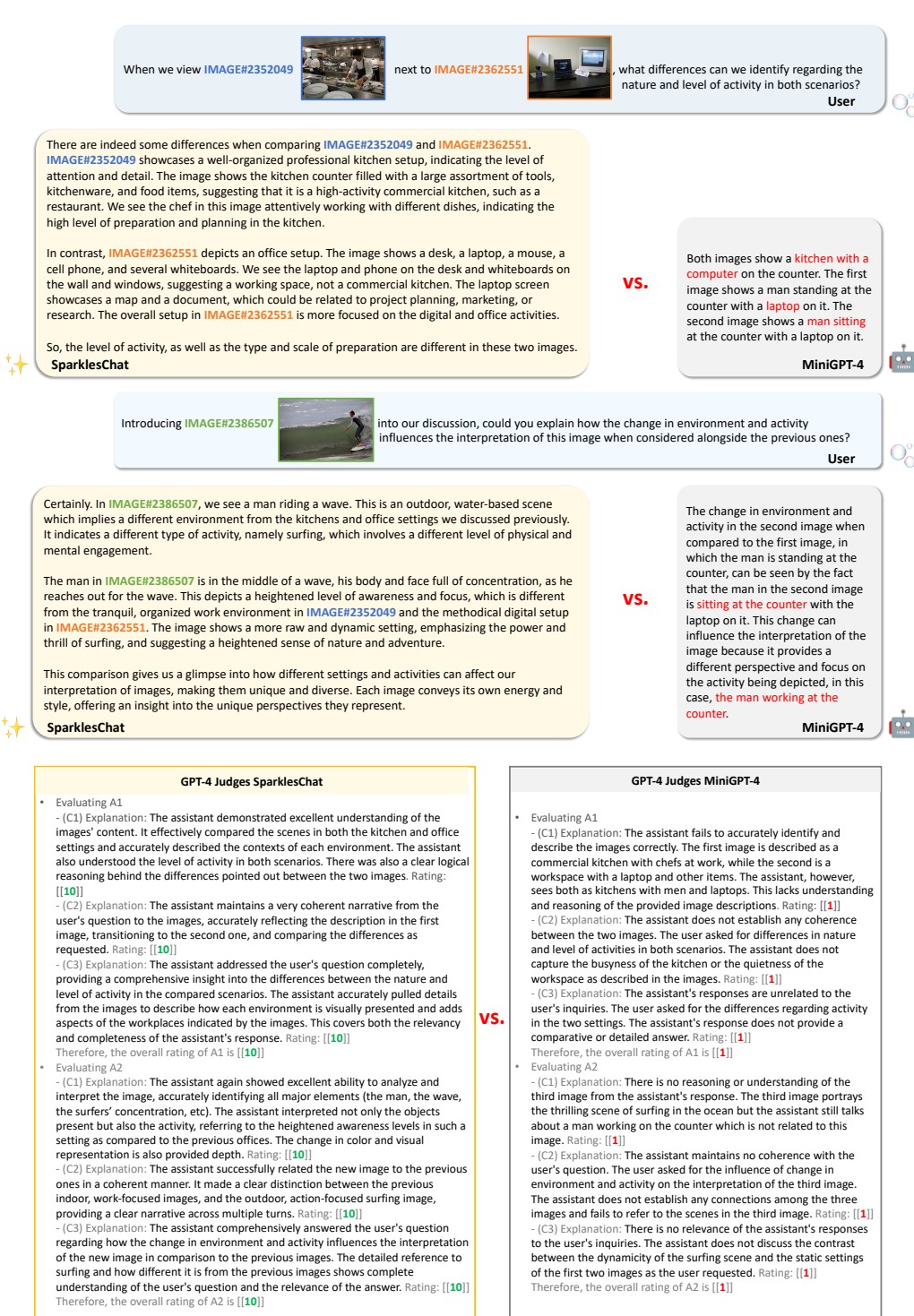

Figure 2: Comparison between our SparklesChat (left) and MiniGPT-4 (right) on an example from SparklesEval. We adapt MiniGPT-4 to accept multiple images as input. SparklesChat shows conversational competence in open dialogues across three criteria: (C1) image understanding and reasoning, (C2) maintaining cross-image and cross-turn coherence, and (C3) generating relevant and complete responses. In contrast, MiniGPT-4 faces challenges in these aspects, leading to difficulty following user instructions across various images and dialogue turns.

2023; Touvron et al., 2023; Chiang et al., 2023; Wei et al., 2022; Wang et al., 2022; Yin et al., 2023a). In the multimodal domain, multimodal instruction-following models such as MiniGPT-4 extend these capabilities by integrating pretrained vision encoders with instruction-following LLMs using projection layers (Zhu et al., 2023a). MiniGPT-4 adapts the projection layer to align vision and language domains by training on concatenated embeddings of images and their descriptions. The training occurs in two stages: first, on a large-scale collection of image-text pairs and then on a smaller dataset of detailed, human-like image descriptions. With this training method, MiniGPT-4 learns alignments between individual images and sentences and performs single-image understanding and reasoning. However, models such as MiniGPT-4 struggle to capture interactions among multiple images and text. This capability is crucial for user-assistant conversations, where users often refer to multiple images with text snippets to convey their instructions in detail. As shown in Figure 2, MiniGPT-4 mixes up the content of multiple images, fails to establish coherence between images, and consequently falls short in following user instructions during open dialogues.

One key limitation hindering progress in this area is the lack of specialized datasets designed for multimodal dialogues that involve multiple images and fine-grained, word-level text interactions. Existing models such as Flamingo can adapt to various image understanding tasks when prompted with a few relevant examples due to their training on image-text interleaved web data (Alayrac et al., 2022). However, these models often fall short in following intricate human instructions because they are trained to predict the next word on a large web dataset rather than perform the task the user wants (Ouyang et al., 2022).

To address these gaps, we present **SparklesChat**, a multimodal instruction-following model for open-ended dialogues across multiple images. Unlike previous approaches such as MiniGPT-4 that takes the concatenation of a single image with sentence-level text as input (e.g., "⭐ Can you describe this image as detailed as possible?" - where '⭐' denotes a single image), SparklesChat, as shown in Figure 1, integrates multiple images at the word level (e.g., "Can you link the celebration occurring in IMAGE#2331159⭐ and the dirt bike race in IMAGE#2330601⭐?"). This innovation enables fine-grained integration of images and text, mimicking natural human communication more closely.

To support the training of SparklesChat, we introduce **SparklesDialogue**, the first machine-generated dialogue dataset designed for word-level interleaved multi-image and text interactions. We use OpenAI's GPT-4 (Openai, 2023) to simulate user-assistant conversations with visual capabilities by leveraging detailed image descriptions. Our dataset achieves greater robustness and diversity by incorporating two subsets, namely SparklesDialogueCC and SparklesDialogueVG, constructed from different image and description sources.

Furthermore, we introduce **SparklesEval**, a GPT-assisted benchmark to quantitatively evaluate a model's conversational competence in multimodal, open-ended dialogues across multiple images and dialogue turns. SparklesEval features a comprehensive and interpretable scoring system based on three distinct criteria: *Image Understanding and Reasoning*, *Cross-Image and Cross-Turn Coherence*, and *Relevance and Completeness of Responses*.

For quantitative evaluation, we validate the effectiveness of SparklesChat through extensive experiments. We conduct zero-shot evaluations on two standard vision-language tasks, including binary image selection on the BISON dataset (Hu et al., 2019) and visual reasoning on the NLVR2 dataset (Suhr et al., 2019). On the BISON dataset, SparklesChat achieved an accuracy of 56.7%, surpassing MiniGPT-4's 46.0%. On the NLVR2 dataset, SparklesChat reached an accuracy of 58.0%, outperforming MiniGPT-4's 51.3%. In our SparklesEval benchmark, SparklesChat scores 8.56 out of 10, significantly exceeds MiniGPT-4's score of 3.91, and closely approaches GPT-4's score of 9.26. Qualitative evaluations further demonstrate SparklesChat's applicability in real-world scenarios.

## 2 RELATED WORKS

Our work in constructing a dialogue dataset for instruction tuning is related to developments in multimodal image-text alignment, dialogue datasets, and instruction tuning. We provide a concise summary of these developments herein and direct readers to Appendix B for a detailed discussion.

**Multimodal alignment datasets.** Various datasets such as Visual Genome (Krishna et al., 2017) and Conceptual Captions (Sharma et al., 2018) align images with corresponding descriptions, form-

ing the foundation for multimodal alignment. Advancements such as the Common Crawl Interleaved data (Huang et al., 2023) and the Multimodal C4 dataset (Zhu et al., 2023b) expand conventional datasets by integrating multiple images and sentences from web corpora. Models including Flamingo (Alayrac et al., 2022) and Kosmos-1 (Huang et al., 2023) trained on them can adapt to various tasks using multiple image-text examples. However, they fall short in following intricate instructions as they are trained to predict the next word on a large web dataset rather than perform the task the user wants (Ouyang et al., 2022).

**Multimodal dialogue datasets.** Datasets such as Visual Dialog (Das et al., 2017), created by crowd workers, and LLaVA (Liu et al., 2023c), generated by LLMs, focus on image-driven conversations inquiring about image attributes or factual knowledge. Conversely, datasets such as OpenViDial (Meng et al., 2020) and PhotoChat (Zang et al., 2021) integrate images within daily human conversations sparsely. Nonetheless, these datasets are not explicitly designed for instructive, in-depth multi-image analysis dialogues, posing challenges in dealing with real-world analytical scenarios.

**Multimodal instruction tuning.** Multimodal instruction tuning developed with datasets like MultiInstruct (Xu et al., 2022) offering benchmarks for diverse multimodal tasks and models like MiniGPT-4 (Zhu et al., 2023a) being fine-tuned on detailed image descriptions to align better with user intentions. Techniques such as LLaVA (Liu et al., 2023c) and SVIT (Zhao et al., 2023) leverage LLMs to interpret image annotations and generate instruction-following datasets. Our dataset, SparklesDialogue, and model, SparklesChat, build upon these developments and explore complex interactions between multiple images and word-level textual content.

## 3 SPARKLESCHAT

We present a multimodal instruction-following model SparklesChat to foster interactions between users and AI assistants across multiple images and illustrate the framework in Figure 1.

**Architecture.** The foundation for SparklesChat is the MiniGPT-4 architecture, which connects a pretrained vision encoder and a pretrained LLM with a projection layer (Zhu et al., 2023a). The language decoder, Vicuna (Chiang et al., 2023), is based on the LLaMA framework (Touvron et al., 2023), which can handle diverse language tasks. For image processing, we use the visual encoder from BLIP-2, combining a pretrained EVA-ViT in Vision Transformer (ViT) backbone with a pretrained Q-Former (Li et al., 2023c; Fang et al., 2022; Dosovitskiy et al., 2021). In the MiniGPT-4, the input to the language model is a single image representation followed by a sentence embedding of the image description. In SparklesChat, image representations of different images are embedded between text according to their positions in dialogues. Only the projection layer is trainable in the model while other vision and language components are frozen. More implementation details are in Appendix C.

**Instruction-tuning.** We simplify the representation of a $T$-turn dialogue $\mathbf{X}^i$ into question-answer pairs for each turn. Training samples are constructed by sequencing these pairs with a predefined system prompt. The prompt $\mathbf{X}^{i,t}_{\text{prompt}}$ and response $\mathbf{X}^{i,t}_{\text{response}}$ at turn $t$ are formulated to incorporate the system prompt and the dialogue content up to that turn. The LLM is trained on this data using an auto-regressive training objective, focusing on predicting the target responses based on the prompts. A more detailed description is available in the Appendix A.

## 4 SPARKLESDIALOGUE AND SPARKLESEVAL

We introduce SparklesDialogue and SparklesEval for training and evaluation to enhance the conversational abilities of multimodal models across multiple images and dialogue turns.

### 4.1 GPT-ASSISTED DATA CONSTRUCTION

We aim to construct a multimodal dialogue dataset that offers fine-grained interactions between multiple images and words, mimicking user-assistant conversations. These dialogues should cover real-world concepts, objects, and entities, spanning scenarios that involve generating text materials,

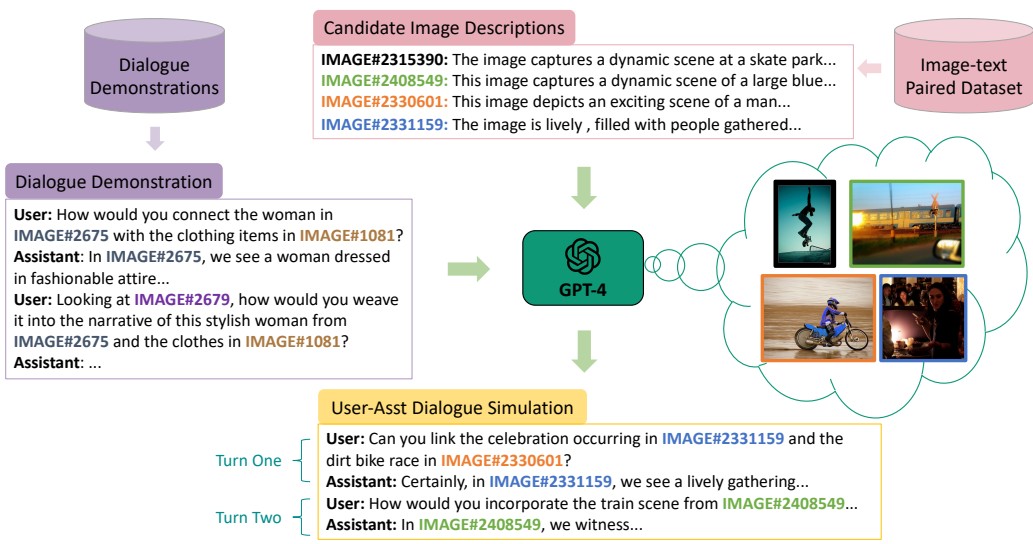

Figure 3: The GPT-assisted data construction process. GPT-4 simulates dialogues between a user and an assistant using multiple images. Dialogue Demonstrations act as learning examples for generating well-formatted dialogues, and Candidate Image Descriptions provide a pool of images for discussion. No visual images are sent to GPT-4 during this process.

seeking advice, guidance, assistance, and much more. To achieve this, we use GPT-4 as the primary tool in our dialogue data generation, given its capabilities to follow complex instructions and extensive world knowledge. The data collection process is visualized in Figure 3. We instruct GPT-4 to simulate realistic and diverse dialogues between a user and an assistant with advanced image understanding and reasoning capabilities following a structured two-turn pattern. In the first turn, the user initiates a reasonable and creative message regarding some images. In response, the assistant generates detailed answers that include comprehensive reasoning regarding the visual content. In the second turn, the user introduces a new image for further discussion, referencing both the new and previous images. Again, we carefully design prompts for the assistant to respond with highly helpful and exceptionally detailed answers that provide comprehensive reasoning to better align with human preference.

`Dialogue Demonstration` and `Candidate Image Descriptions` are crucial components in this process. `Dialogue Demonstrations` serve as in-context learning examples, steering GPT-4 towards generating well-formatted and diverse responses. We showcase and analyze `Dialogue Demonstrations`' contribution to data quality and diversity in Appendix K. We initiated the creation of hundreds of demonstration dialogues with GPT-4's assistance, using similar prompts and checking their quality. A small subset of them is randomly chosen each time. `Candidate Image Descriptions` serves as a candidate pool for relevant image selection. From the pool of image-text paired dataset, we randomly select a small subset as candidates each time. We include image IDs in dialogues to avoid reference ambiguity. Given that the publicly accessible GPT-4 API only accepts text input, we represent images with detailed descriptions. These descriptions, sourced from various image annotations such as image captions, bounding boxes, and region descriptions, comprehensively portray image content (Zhu et al., 2023a; Zhao et al., 2023; Liu et al., 2023c). We parse and retain well-structured responses that conform to our desired format. More details, such as prompt templates and visualized examples, are in Appendix J.

## 4.2 STATISTICS AND CHARACTERISTICS

We use the data construction process introduced in subsection 4.1 to collect a robust and diverse dataset named SparklesDialogue, which consists of two subsets: SparklesDialogueCC and Sparkles-DialogueVG. The respective detailed descriptions, provided in MiniGPT-4 (Zhu et al., 2023a) and SVIT (Zhao et al., 2023), correspond to image sources from Conceptual Captions (CC) (Sharma et al., 2018) and Visual Genome (VG) (Krishna et al., 2017). SparklesDialogueVG is of high quality as the VG image descriptions generated by GPT-4 benefit from human-annotated captions, objects,

Table 1: Statistics of SparklesDialogue and SparklesEval.

| Dataset Name | Image Source | Caption Source | #Dialogue | #Image Turn one | #Image Turn two | #Unique/Total Image | |
|---|---|---|---|---|---|---|---|
| SparklesDialogueCC | CC | MiniGPT-4 | 1,653 | 1 | 1 | 2,067/3,306 | |
| | | | 1,799 | 2 | 1 | 2,642/5,397 | 3,373/12,979 |
| | | | 1,069 | 3 | 1 | 2,408/4,276 | |
| SparklesDialogueVG | VG | SVIT | 1,000 | 2 | 1 | 3,000/3,000 | 7,000/7,000 |
| | | | 1,000 | 3 | 1 | 4,000/4,000 | |
| SparklesEval | VG | SVIT | 50 | 2 | 1 | 150/150 | |
| | | | 50 | 2 | 2 | 200/200 | 550/550 |
| | | | 50 | 3 | 1 | 200/200 | |

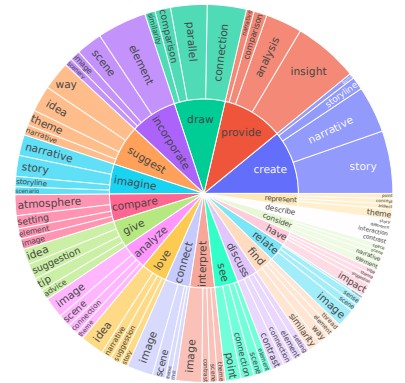

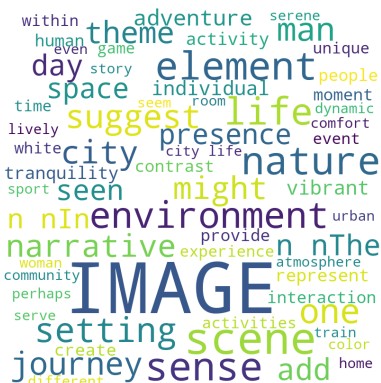

(a) Root verb-noun pairs in user messages.

(b) Word cloud of assistant messages.

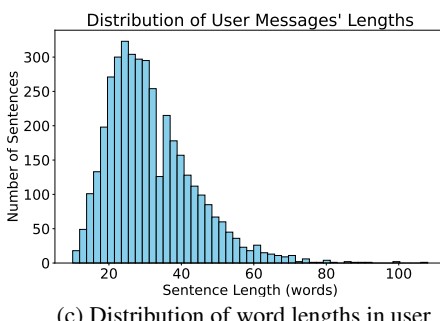

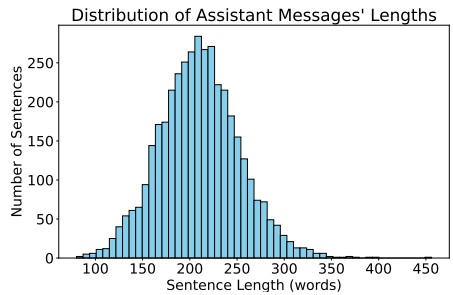

(c) Distribution of word lengths in user messages with an average of 31.5 words.

(d) Distribution of word lengths in assistant messages with an average of 210.5 words.

Figure 4: Characteristics of SparklesDialogueVG.

and regions (Zhao et al., 2023). On the other hand, SparklesDialogueCC enriches SparklesDialogue by drawing from a more extensive set of images – 3.3 million in CC compared to 0.1 million in VG. However, the CC image descriptions are generated by a multimodal model with image features but not human annotations and are more prone to object hallucination issues (Zhu et al., 2023a). Our ablation study elaborated in section 5.2 demonstrates that combining these two subsets improves SparklesChat's capacity for understanding and reasoning across images and text. SparklesEval emphasizes more on accuracy and is thus constructed using the same source as SparklesDialogueVG. To encourage diversity in evaluation, SparklesEval was curated by analyzing the verb-noun distribution in user questions and selecting those that appear only once.

Table 1 provides the data statistics for SparklesDialogue and SparklesEval. SparklesDialogueCC comprises 4.5K dialogues, each consisting of at least two images spanning two conversational turns. On the other hand, SparklesDialogueVG includes 2K dialogues, each with at least three distinct images across two turns. SparklesEval includes 150 dialogues, with one-third containing two images in both the first and second conversational turns.

Figure 4 shows the characteristics of our dataset using SparklesDialogueVG as a representative subset. We explore key elements such as the root verb-noun pairs in user messages, a word cloud of assistant messages, and the length distributions. The questions from users are diverse, ranging from generating text materials to seeking advice or discussing the relationships between images, such as comparison and connection. The dialogues span various real-world topics, including the environment, nature, life, cities, etc. The high average word count in assistant messages suggests that the responses in SparklesDialogue are thorough and detailed. For details on the characteristics visualization of SparklesDialogueCC and extracting root verb-noun pairs and their visualization based on image count in each turn, please refer to Appendix H.

### 4.3 GPT-ASSISTED EVALUATION: SPARKLESEVAL

While previous research, such as visual storytelling, has leaned toward human evaluations as superior to quantitative measures, these evaluations are often subjective, costly, and time-consuming (Huang et al., 2016). Inspired by the consistency of recent LLMs with human assessment in evaluating output quality (Zheng et al., 2023), we developed SparklesEval, a GPT-assisted benchmark to quantitatively assess a model's conversational competence across multiple images and dialogue turns.

Each dialogue under evaluation comprises two turns of questions from SparklesEval and model-generated answers, considering both the current question and any preceding dialogue history. To evaluate a dialogue, we provide the judge model (i.e., GPT-4) with the complete dialogue but only require the judge model to assess the two answers. Descriptions corresponding to images mentioned in the dialogue are also provided. Each assessment is based on three distinct criteria across two turns, providing reasons and ratings on a scale of 1 to 10. These criteria scores include C1 for Image Understanding and Reasoning, C2 for Cross-Image and Cross-Turn Coherence, and C3 for Relevance and Completeness of Responses, with details in Appendix D. In particular, we further introduce averaged scores **A1** and **A2** for each turn, and an average **Score** computed by averaging **A1** and **A2**. The evaluation prompt and score computation process are elaborated in Appendix D.

Our evaluation approach differs from prior GPT-assisted evaluations in two aspects. First, it employs a combined score for a more comprehensive and interpretable assessment instead of a singular one (Liu et al., 2023c). Second, it is less biased and more efficient by assessing a single dialogue per prompt rather than contrasting multiple dialogues within one prompt (Zheng et al., 2023). Our approach eliminates position bias - the potential favor to certain positions when multiple dialogues are assessed within a prompt (Zheng et al., 2023). It enhances efficiency by avoiding the recalculation of combined scores for multiple dialogues.

## 5 EXPERIMENTS

Beyond assessing conversational competence using SparklesEval, we evaluate models' zero-shot understanding and reasoning capabilities across multiple images through binary image selection on BISON and visual reasoning with natural language on NLVR2 (Hu et al., 2019; Suhr et al., 2019).

### 5.1 COMPARISON OF MODEL PERFORMANCE

Table 2 compares the performance of SparklesChat, MiniGPT-4, and LLaVA-v1.5 (Liu et al., 2023b) on BISON, NLVR2, and SparklesEval evaluation datasets. When SparklesChat is trained on reasoning data adapted from LLaVA (Liu et al., 2023c), it achieves improved performance over models trained on description data on all metrics. This emphasizes the importance of reasoning ability.

SparklesChat, trained on our SparklesDialogue, outperforms other multimodal models in three tasks. SparklesChat achieves accuracies of 56.7% and 58.0% on BISON and NLVR2, respectively, reflecting its efficacy in handling tasks that require fine-grained visual grounding and compositional visual reasoning over two images. Moreover, SparklesChat significantly excels in the SparklesEval benchmark, scoring 8.56 out of 10. Its highest scores in both the first and second turns across all criteria indicate its superior ability in image understanding and reasoning, maintaining cross-image and cross-turn coherence, and generating relevant and complete responses. In comparison, models trained on description and reasoning data approximate scores of 3 and 6.71, respectively. GPT-4 scores the highest at 9.26, largely attributed to its utilization of detailed ground-truth annotations.

Table 2: Model comparison on BISON, NLVR2 and SparklesEval. We investigate training models on different data sources, including detailed descriptions, complex reasoning, and dialogue data.

| Model | Instruction Data | BISON | NLVR2 | SparklesEval | | | | | | | | |
| | | | | Score | Turn one | | | | Turn two | | | |
| | | | | | A1 | C1 | C2 | C3 | A2 | C1 | C2 | C3 |
| GPT-4 | - | - | - | 9.26 | 9.26 | 9.23 | 9.18 | 9.38 | 9.26 | 9.25 | 9.15 | 9.38 |
| MiniGPT-4 | description | 46.0% | 51.3% | 3.91 | 3.55 | 3.67 | 3.53 | 3.44 | 4.28 | 4.38 | 4.21 | 4.23 |
| MiniGPT-4* | description | 51.3% | 46.7% | 3.50 | 3.14 | 3.27 | 3.16 | 3.00 | 3.86 | 3.97 | 3.81 | 3.81 |
| LLaVA-v1.5 | Mixture (665K) | 52.7% | 53.3% | 2.75 | 2.80 | 2.74 | 2.94 | 2.71 | 2.69 | 2.69 | 2.70 | 2.68 |
| LLaVA-v1.5* | +SparklesDialogue | **65.3%** | 56.7% | 7.93 | 7.54 | 7.37 | 7.73 | 7.51 | 8.32 | 8.21 | **8.36** | 8.39 |
| SparklesChat | description | 52.0% | 48.0% | 3.06 | 2.64 | 2.79 | 2.67 | 2.46 | 3.48 | 3.76 | 3.40 | 3.29 |
| SparklesChat | reasoning | 52.7% | 54.0% | 6.71 | 6.55 | 6.63 | 6.42 | 6.59 | 6.87 | 6.89 | 6.73 | 6.98 |
| **SparklesChat** | **SparklesDialogue** | 56.7% | **58.0%** | **8.56** | **8.76** | **8.81** | **8.67** | **8.81** | 8.35 | 8.37 | 8.28 | **8.41** |

The evaluation metric is accuracy for BISON and NLVR2; SparklesEval is rated 1-10. All models are adapted to accept multiple images for fair comparison. MiniGPT-4* is our reimplementation of MiniGPT-4 with the same setup as SparklesChat. Description and reasoning datasets from LLaVA are adapted using formats similar to SparklesDialogue, with overlapping samples removed between train and evaluation sets. LLaVA-v1.5 has improved its image input resolution from 224 pixels to 336 pixels. It is trained on a diverse dataset comprising about 665K instruction-following data. Additionally, LLaVA-v1.5* is fine-tuned on SparklesDialogue (about 6.5K) using the low-resource technique LoRA.

SparklesChat's score is about 92% of that of GPT-4, highlighting SparklesChat's conversational competence across various images and dialogue turns.

SparklesChat is built upon the MiniGPT-4. To offer a broader view of our approach's effectiveness, we extended our approach to more advanced models such as LLaVA-v1.5 (Liu et al., 2023c;b). Despite LLaVA-v1.5's advantages of higher image resolution (336 vs. 224 pixels) and a larger training set (665K vs. 6.5K), SparklesChat significantly outperforms LLaVA-v1.5 in three evaluation sets involving multiple images. While LLaVA-v1.5 outperforms MiniGPT-4 on BISON and NLVR2, it shows weaker results on SparklesEval. This may be due to LLaVA-v1.5's training data primarily focusing on closed-set multimodal tasks such as VQA, TextCaps, and RefCOCO, while lacking in open-ended dialogue training. After fine-tuning with our SparklesDialogue, LLaVA-v1.5* not only significantly improved in open-ended dialogue tasks but enhanced traditional multimodal tasks. These results validate the adaptability of our method in unlocking chats across multiple images for multimodal instruction-following models with minimal additional training cost.

For a comprehensive understanding, refer to Appendix E for evaluation settings including evaluation protocol and prompt design. Side-by-side comparisons of example outputs for SparklesChat and MiniGPT-4 on SparklesEval, BISON, and NLVR2 are illustrated in Figure 2 and Appendix E. Additionally, Appendix G provides a detailed evaluation of GPT-4, MiniGPT-4, and SparklesChat on SparklesEval using three different versions of judge models.

## 5.2 ABLATION STUDIES

We study the effect of training SparklesChat using data variants of SparklesDialogue on dialogue turn ratios and different subsets and show results in Table 3.

**Effect of dialogue turns in SparklesDialogue.** We first train models with individual dialogue turns. The model trained solely on the first turn (turns ratio '1:0') performs better across all metrics compared to the model solely trained on the second turn (turns ratio '0:1'). Additionally, this model outperforms those trained on the baseline datasets, as evidenced in Table 2. This illustrates that our dataset boosts reasoning and conversational abilities even when used in isolation with just the first turn. In contrast, training only with the second dialogue turn reduces scores on BISON and NLVR2 when compared with baselines in Table 2. This could stem from the extended prompts in the second turn, which includes the content of the first turn, making them less aligned with the short prompt format favored by BISON and NLVR2. Then, we train models with SparklesDialogue blending with the same ratios of samples constructed from two dialogue turns, represented by turns ratio '1:1' in Table 2. The results are better than only training with the second turn in all metrics, while worse

Table 3: Ablation studies on BISON, NLVR2, and SparklesEval analyzing the effects of training SparklesChat with variations of SparklesDialogue on dialogue turn ratios and different subsets.

| Data | Turns Ratio | BISON | NLVR2 | SparklesEval | | | | | | | | |
|------|-------------|-------|-------|-------|------|------|------|------|------|------|------|------|
| | | | | Score | Turn one | | | | Turn two | | | |
| | | | | | A1 | C1 | C2 | C3 | A2 | C1 | C2 | C3 |
| SparklesDialogue (CC+VG) | 1:0 | 57.3% | 55.3% | 8.50 | 8.65 | 8.70 | 8.52 | 8.73 | 8.35 | 8.38 | 8.24 | 8.44 |
| | 0:1 | 50.7% | 46.7% | 8.24 | 8.24 | 8.23 | 8.18 | 8.32 | 8.24 | 8.24 | 8.15 | 8.32 |
| | 1:1 | 59.3% | 51.3% | 8.73 | 8.80 | 8.83 | 8.66 | 8.91 | 8.65 | 8.62 | 8.55 | 8.79 |
| | 1:2 | 49.3% | 51.3% | 8.43 | 8.54 | 8.57 | 8.43 | 8.63 | 8.31 | 8.28 | 8.21 | 8.43 |
| | 2:1 | 56.7% | 58.0% | 8.56 | 8.76 | 8.81 | 8.67 | 8.81 | 8.35 | 8.37 | 8.28 | 8.41 |
| | 3:1 | 50.7% | 48.7% | 8.45 | 8.69 | 8.74 | 8.52 | 8.83 | 8.20 | 8.18 | 8.08 | 8.33 |
| SparklesDialogueCC | 2:1 | 44.7% | 53.3% | 8.18 | 8.26 | 8.29 | 8.16 | 8.33 | 8.10 | 8.10 | 8.00 | 8.20 |
| SparklesDialogueVG | 2:1 | 54.7% | 52.0% | 8.59 | 8.71 | 8.76 | 8.60 | 8.78 | 8.47 | 8.47 | 8.35 | 8.60 |

The evaluation metric is accuracy for BISON and NLVR2; SparklesEval is rated 1-10.

than only training from the first turn in the task of NLVR2. An increase in the sampling ratios of the second turn data (turns ratio '1:2') results in a performance drop as expected. Therefore, we increase the sampling ratio of the first-turn data until we cannot observe performance boosting. We finally settled on a 2:1 ratio for the first turn to the second turn as our default setting as it achieves balanced good performance across all benchmarks.

**Effect of subsets of SparklesDialogue.** Our model has been trained on two subsets of SparklesDialogue: SparklesDialogueCC and SparklesDialogueVG. The model trained on SparklesDialogueVG outperforms that trained on SparklesDialogueCC in both the BISON and SparklesEval evaluations, scoring 54.7% and 8.59, respectively, compared to 44.7% and 8.18. This enhanced performance is partly due to the higher quality of SparklesDialogueVG, which benefits from human-annotated data as discussed in subsection 4.2. It is worth noting that SparklesDialogueVG and SparklesEval use the same sources of images and captions, which could partially account for the higher score achieved by SparklesDialogueVG on SparklesEval. Both subsets demonstrate similar efficacy on the NLVR2 test. Combining both subsets yields higher performance on the BISON and NLVR2 tests, scoring 56.7% and 58.0% respectively. This surpasses the scores achieved by using either subset alone. In addition, the model trained on the combined dataset performs comparably to SparklesDialogueVG in the SparklesEval test, scoring 8.56 versus 8.59. This suggests that combining SparklesDialogueVG's high-quality data and SparklesDialogueCC's diverse data results in a more robust and versatile dataset for enhancing models' capabilities in understanding and reasoning across images and text.

## 5.3 DEMONSTRATIONS AND APPLICATIONS

We conducted qualitative demonstrations to showcase SparklesChat's broad applications in free-form scenarios to handle dialogues involving two or three-turn dialogues, with each turn involving images from one to five. The visualization and analysis of results are shown in Appendix F.

## 6 CONCLUSION

**In conclusion**, this work unlocks multimodal instruction-following models' capabilities in open-ended dialogues involving multiple images. We introduced SparklesChat, a model designed to handle word-level text interactions in a multimodal context, offering natural conversational flow and direct context awareness. We also presented SparklesDialogue, the first machine-generated dialogue dataset tailored for multi-image and word-level text interactions. Furthermore, we proposed SparklesEval, a specialized benchmark for quantitatively assessing a model's multimodal conversational competence. Experimental results demonstrated SparklesChat's superiority over existing models in both standard vision-language tasks and the newly-introduced SparklesEval benchmark. We also conducted qualitative demonstrations to showcase the model's broad applications in free-form scenarios. We discuss **limitations** of this work to inspire **future research** in this field in Appendix I. We have uploaded the code, model, data, and detailed instructions as supplementary material to ensure **reproducibility**.

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

# Appendix

## A  INSTRUCTION-TUNING DETAILS OF SPARKLESCHAT

We represent an $i$-th $T$-turn dialogue as $\mathbf{X}^i = \left(\mathbf{X}_{\mathsf{q}}^{i,1}, \mathbf{X}_{\mathsf{a}}^{i,1}, \cdots, \mathbf{X}_{\mathsf{q}}^{i,T}, \mathbf{X}_{\mathsf{a}}^{i,T}\right)$, where each pair of $\left(\mathbf{X}_{\mathsf{q}}^{i,t}, \mathbf{X}_{\mathsf{a}}^{i,t}\right)$ includes a question from the user and an answer from the assistant in turn-$t$. For each $\mathbf{X}^i$, we construct $T$ training samples by organizing each pair of questions and answers as a sequence. Given a predefined system prompt $\mathbf{X}_{\mathsf{system}}$, the prompt $\mathbf{X}_{\mathsf{prompt}}^{i,t}$ and response $\mathbf{X}_{\mathsf{response}}^{i,t}$ at the $t$-th turn are defined as the following:

$$\mathbf{X}_{\mathsf{prompt}}^{i,t} = \begin{cases} \mathbf{X}_{\mathsf{system}}\texttt{<SEP>Human}: \mathbf{X}_{\mathsf{q}}^{i,1}\texttt{<SEP>Assistant}:, & \text{if } t = 1, \\ \mathbf{X}_{\mathsf{prompt}}^{i,t-1}\texttt{<SEP>Human}: \mathbf{X}_{\mathsf{q}}^{i,t}\texttt{<SEP>Assistant}:, & \text{if } t > 1. \end{cases} \tag{1}$$

$$\mathbf{X}_{\mathsf{response}}^{i,t} = \mathbf{X}_{\mathsf{a}}^{i,t}\texttt{<SEP>}. \tag{2}$$

We train the LLM on the prediction tokens using the auto-regressive training objective. Specifically, for a sequence of length $L$, we compute the probability of generating target responses $\mathbf{X}_{\mathsf{response}}$ by:

$$p\left(\mathbf{X}_{\mathsf{response}}|\mathbf{X}_{\mathsf{prompt}}\right) = \prod_{l=1}^{L} p_{\boldsymbol{\theta}}\left(\boldsymbol{x}_l|\mathbf{X}_{\mathsf{prompt},<l}, \mathbf{X}_{\mathsf{response},<l}\right), \tag{3}$$

where $\boldsymbol{\theta}$ is the trainable parameters, $\mathbf{X}_{\mathsf{prompt},<l}$ and $\mathbf{X}_{\mathsf{response},<l}$ are prompt and response tokens in all turns before the current prediction token $\boldsymbol{x}_l$, respectively.

## B  RELATED WORKS

Our work exploits image-text pairs to construct a dialogue dataset for instruction-tuning. Thus, we review related works on multimodal alignment datasets, multimodal dialogue datasets, and multimodal instruction tuning, primarily on natural images and text domains.

**Multimodal alignment datasets.** Various datasets, such as MSCOCO (Lin et al., 2014), Visual Genome (Krishna et al., 2017), Conceptual Captions (Sharma et al., 2018), Conceptual 12M (Changpinyo et al., 2021), ALIGN (Jia et al., 2021) and LAION (Schuhmann et al., 2021), have been constructed to **align images with their corresponding descriptions**. These datasets have significantly contributed to the development of multimodal models for image-and-text generation (Huang et al., 2021a; Rombach et al., 2022; Li et al., 2023c) and understanding (Jia et al., 2021; Huang et al., 2021b; Radford et al., 2021). We use these datasets in our data construction process. Emerging trends include datasets featuring **interleaved images and text sequences from web corpora**, such as M3W (Alayrac et al., 2022), web and Wikipedia articles (Aghajanyan et al., 2022), Common Crawl Interleaved data (Huang et al., 2023), and the Multimodal C4 dataset (Zhu et al., 2023b). These datasets extend conventional image-text alignment training by incorporating multiple images and sentences. When trained on these enriched datasets, models such as Flamingo (Alayrac et al., 2022), OpenFlamingo (Awadalla et al., 2023), Kosmos-1 (Huang et al., 2023), and EMU (Sun et al., 2023) can adapt to various image understanding tasks using multiple task-relevant image-text examples. However, these models often fall short in following intricate human instructions because they are trained to predict the next word on a large web dataset rather than perform the task the user wants (Ouyang et al., 2022).

**Multimodal dialogue datasets.** Existing multimodal dialogue datasets broadly fall into two categories. The first comprises datasets where **conversations are heavily rooted in and driven by images**. Traditional datasets of this type are primarily generated by inviting crowd workers to engage in dialogues about a common image. Notable examples include Visual Dialog (Das et al., 2017), which emphasizes question-answering tasks within AI-human chat about visual content, and IGC (Mostafazadeh et al., 2017), a compilation of dialogues featuring an image, a corresponding textual description, and a conversation centered on the image. Image-Chat presents image-grounded dialogues crafted around given images (Shuster et al., 2020). Recently, dialogue datasets, such as LLaVA (Liu et al., 2023c), SVIT (Zhao et al., 2023), and LAMM (Yin et al., 2023b), created by LLMs alongside image annotations have surfaced. Each dialogue in these datasets begins with an inquiry about image attributes or factual knowledge, with responses expected to be brief within 50 words, which may not align with real-world scenarios requiring in-depth multi-image analysis. The second category features **daily human conversations**, with images interspersed within multi-turn conversations sparsely. For example, OpenViDial (Meng et al., 2020; Wang et al., 2021) is sourced from dialogues in movies and TV series, whereas PhotoChat (Zang et al., 2021) is a human-human dialogue dataset developed through crowdsourcing and features photo-sharing. Other datasets, such as DialogCC (Lee et al., 2022), MultiModalDialogue (Lee et al., 2021), and IMAD (Viktor & Denis, 2023) enhance text-only dialogues by incorporating semantically relevant images. In addition, MM-Chat (Zheng et al., 2022) and MMDialog (Feng et al., 2023) encompass image-grounded dialogues derived from social media interactions. However, these datasets, not being designed for user-assistant interactions, struggle with instructive, problem-solving dialogue requirements.

**Multimodal instruction tuning.** Multimodal instruction tuning has grown substantially with the advent of multimodal instruction datasets. For instance, MultiInstruct (Xu et al., 2022) offers a benchmark comprising 62 diverse multimodal tasks unified in a seq-to-seq format. InstructBLIP (Dai et al., 2023) extended the scope by transforming 26 datasets into instruction-tuning form. Otter (Li et al., 2023b) is trained on MIMIC-IT (Li et al., 2023a), a multimodal in-context instruction tuning dataset constructed by grouping multiple similar instructions into a contextual example. To better align with user intentions, MiniGPT-4 is fine-tuned on a small dataset of detailed image descriptions (Zhu et al., 2023a) and PF-1M (Chen et al., 2023) rewrites image annotations in a human-like style across 37 vision-language datasets. Furthermore, techniques such as LLaVA (Liu et al., 2023c), SVIT (Zhao et al., 2023), LRV-Instruction (Liu et al., 2023a), and LAMM (Yin et al., 2023b) have emerged. These methods leverage language-only APIs such as OpenAI's GPT-4 (Openai, 2023) and self-instruction methods (Wang et al., 2022) to interpret image annotations (e.g., image captions, region descriptions, object bounding boxes, attributes, and relationships), and generate responses in various forms (i.e., short conversations, image captioning, and visual reasoning). Models such as mPLUG-Owl (Ye et al., 2023), PandaGPT (Su et al., 2023), LLaMAAdapter V2 (Gao et al., 2023), and Multimodal-GPT (Gong et al., 2023) further extended this area, incorporating both language-only and vision-language instruction data. These developments are a valuable foundation for our work. Our dataset, SparklesDialogue, is inspired by GPT-assisted data construction techniques and explores the interactions between multiple images and word-level textual content. Training our model,

SparklesChat, on this dataset unlocks the capability of multimodal models to interpret complex image-text interactions.

## C  IMPLEMENTATION DETAILS

We implemented SparklesChat on the MiniGPT-4 codebase (Zhu et al., 2023a)[1]. We refer to MiniGPT-4's efficient fine-tuning process and tune SparklesChat using 1,500 training steps with a batch size of 8, based on MiniGPT-4's first-stage pretrained model. Our training data of SparklesDialogue is sampled with the same ratio from SparklesDialogueCC and SparklesDialogueVG, and with sampling ratios of 2 and 1 from the first and second turns of dialogues, respectively. When comparing our model with LLaVA-v1.5, we utilized the code from the LLaVA codebase[2] (Liu et al., 2023c;b). For fine-tuning LLaVA-v1.5* on SparklesDialogue, we adopted a learning rate of 2e-5 and a batch size of 4 per GPU. The training was conducted over 2,000 steps across four GPUs, employing the low-resource technique LoRA to save memory and time. The language model components of all models are based on the 7-billion parameter size of the Vicuna architecture (Chiang et al., 2023).

In our evaluation, we configure the parameters as follows: `temperature` is set to 1.0, `top_p` to 0.9, and `max_new_tokens` to 300, with both `repetition_penalty` and `length_penalty` at 1.0. For demonstration cases, the `beam_size` is 2; for all other evaluations, it is 1.

During instruction-tuning, we follow MiniGPT-4 to use `<ImageHere></Img>` to represent images (Zhu et al., 2023a). In practice, all tags of `<ImageHere>` are replaced by the visual features produced by a linear projection layer. Tags of `` and `</Img>` are language tokens that serve as signals for the start and end of images. A system message $\mathbf{X}_{\text{system}}$ is appended to the beginning of each prompt. We also append `Human:` and `Assistant:` before each user and assistant messages to equip the model with conversation capability. System, user, and assistant messages are separated by a separator `<SEP>`. The system message $\mathbf{X}_{\text{system}}$ = `Give the following image: ImageContent</Img>. You will be able to see the image once I provide it to you. Please answer my questions.` The separator `<SEP>` = `###`. Table 4 illustrates the unified format for two-turn dialogue training sequences.

Table 4: Prompt and response sequence formats used to train SparklesChat. The first and the second conversation turns are illustrated here. The model is trained to predict the assistant answers, and thus only green sequence are used to compute the loss in the auto-regressive model. We do not compute the regression loss for the prompt $\mathbf{X}_{\text{prompt}}$ since the prompt is provided by users in real-world applications, making it unnecessary for the model to make predictions in this context.

**Dialogue Turn One**
$\mathbf{X}_{\text{prompt}}^{i,1}$ = $\mathbf{X}_{\text{system}}$`<SEP>Human:` $\mathbf{X}_{\text{q}}^{i,1}$`<SEP>Assistant:`
$\mathbf{X}_{\text{response}}^{i,1}$ = $\mathbf{X}_{\text{a}}^{i,1}$`<SEP>`

**Dialogue Turn Two**
$\mathbf{X}_{\text{prompt}}^{i,2}$ = $\mathbf{X}_{\text{system}}$ `<SEP> Human:` $\mathbf{X}_{\text{q}}^{i,1}$`<SEP> Assistant:` $\mathbf{X}_{\text{a}}^{i,1}$`<SEP> Human:` $\mathbf{X}_{\text{q}}^{i,2}$`<SEP> Assistant:`
$\mathbf{X}_{\text{response}}^{i,2}$ = $\mathbf{X}_{\text{a}}^{i,2}$`<SEP>`

We tailored the OpenAI's GPT-4 API (`gpt-4-0613`) parameters to balance diversity and quality for constructing SparklesDialogue and SparklesEval. We set the `temperature` and `top_p` parameters to 1.0, the `max_tokens` parameter to 2,048, and both the `frequency_penalty` and `presence_penalty` parameters to 0.0. In each query to the GPT-4 API, the "system" role was allocated the default instruction `You are a helpful assistant.` As of July 2023, the cost for generating 1,000 tokens was $0.06 for outputs and $0.03 for inputs within an 8K context[3], leading to a total dataset generation cost of approximately $500. The cost of evaluating a model on

---

[1] `https://github.com/Vision-CAIR/MiniGPT-4`
[2] `https://github.com/haotian-liu/LLaVA/`
[3] `https://openai.com/pricing`

SparklesEval is approximately \$1.4 and \$14 using `gpt-3.5-turbo-0613` and `gpt-4-0613`, respectively.

## D  SPARKLESEVAL DETAILS

The three criteria of GPT-assisted evaluation on SparklesEval are as follows:

**Image understanding and reasoning score** C1: Assess the assistant's proficiency in accurately identifying and describing objects, contexts, and relationships within and across the images.
**Cross-image and cross-turn coherence score** C2: Evaluate the assistant's ability to maintain consistent understanding across multiple images and dialogue turns.
**Relevance and completeness of responses score** C3: Determine the extent to which the assistant's responses are directly related to the user's inquiries and the images' content, and whether the responses provide comprehensive and detailed answers.

Following this, we ask the GPT models to assign a combined score for each turn. For each model's evaluation results, we gather scores for three criteria across two turns. First, we compute the mean scores for all criteria over evaluation samples. Next, we calculate the combined scores **A1** and **A2** by averaging their respective criteria scores, namely $A1 = mean(C1, C2, C3)$ for the first turn and $A2 = mean(C1, C2, C3)$ for the second turn. We refrain from using the **A1** and **A2** scores provided by the judge models, as their calculations may be inaccurate. Ultimately, we derive a final overall **score** by averaging **A1** and **A2**. Through this methodology, our evaluation is more holistic and interpretable.

The prompt template of GPT-assisted evaluation on SparklesEval is presented in Table 5.

## E  ZERO-SHOT EVALUATION ON VISION-LANGUAGE TASKS

We chose two vision-language tasks, binary image selection and visual reasoning, to evaluate zero-shot understanding and reasoning capabilities over multiple images. For both tasks, the evaluation metric is accuracy. Side-by-side comparisons of example outputs for SparklesChat and MiniGPT-4 on BISON and NLVR2 can be found in Figure 5.

**Binary image selection on BISON.**    The Binary Image Selection task measures a model's ability to select the correct image from a pair given a text query that describes one of them (Hu et al., 2019). The model's performance is assessed in terms of binary classification accuracy. For this task, 150 examples were randomly sampled from the COCO-BISON dataset[4]. The image source of COCO-BISON is COCO images. The image source of SparklesDialogueCC is Conceptual Captions, which should have no overlap with COCO. However, our SparklesDialogueVG originates from the Visual Genome, which includes a subset of COCO images. We carefully eliminate any overlapping images to ensure no overlap between the training and evaluation data.

**Visual reasoning with natural language on NLVR2.**    The evaluation of the Visual Reasoning with Natural Language task assesses the model's ability to predict whether a sentence is true about a pair of images (Suhr et al., 2019). This task addresses the challenge of compositional visual reasoning on relations, comparisons, and quantities. The NLVR2 dataset (Suhr et al., 2019) was used for this evaluation, with 150 examples randomly sampled from the public balanced test set[5]. The images in the NLVR2 dataset are sourced from Google Images, distinct from our SparklesDialogueVG's image source of the Visual Genome (Krishna et al., 2017) and primarily feature images from Flickr.

**Evaluation protocol and prompt design.**    Models are evaluated on these tasks without any additional training. Inspired by (Kojima et al., 2022), we used a simple prompt, *"Let's think step by step"*, to facilitate step-by-step reasoning before answering each question. We used the phrase *"Therefore, the answer is"* to prompt the answer. Instead of using a two-stage prompting as in (Kojima et al.,

---

[4]`https://github.com/facebookresearch/binary-image-selection/blob/main/annotations/bison_annotations.cocoval2014.json`
[5]`https://github.com/lil-lab/nlvr/blob/master/nlvr2/data/balanced/balanced_test1.json`

Table 5: Prompt format for SparklesEval evaluation.

Users will interact with a conversational assistant. The assistant is designed to understand, analyze, and reason about multiple images across two turns of conversation. The assistant is expected to provide highly helpful and exceptionally detailed answers providing comprehensive reasoning regarding the visual content of the images.

Below are images represented by their image IDs and captions (delimited by triple quotes):

```json
{Target Image Descriptions}
```

Next is a dialogue between a user and the assistant regarding the images above:
```

###User Q1:
{Q1}

###Assistant A1:
{A1}

###User Q2:
{Q2}

###Assistant A2:
{A2}
```

Your task as an impartial judge is to evaluate the responses (A1 and A2) provided by the assistant to the user's questions.
Please rate the following three criteria C1, C2, and C3 on a scale of 1-10 for A1 and A2 separately, where a higher score indicates better overall performance:
(C1) Image Understanding and Reasoning: This measures the assistant's ability to accurately identify and describe objects, context, and relationships within and between the images.
(C2) Cross-Image and Cross-Turn Coherence: This evaluates the assistant's ability to maintain a consistent understanding across multiple images and dialogue turns.
(C3) Relevance and Completeness of Responses: This assesses whether the assistant's responses are directly related to the user's inquiries and the images' content, and whether the responses provide thorough, detailed answers.

Begin your evaluation by providing a short explanation for each criterion. Be as objective as possible. After providing your explanation, rate the response on a scale of 1 to 10 by strictly following the format below (note that "5" and "..." are placeholders):
```

* Evaluating A1
- (C1) Explanation: "..." Rating: [[5]]
- (C2) Explanation: "..." Rating: [[5]]
- (C3) Explanation: "..." Rating: [[5]]
Therefore, the overall rating of A1 is [[5]]

* Evaluating A2
- (C1) Explanation: "..." Rating: [[5]]
- (C2) Explanation: "..." Rating: [[5]]
- (C3) Explanation: "..." Rating: [[5]]
Therefore, the overall rating of A2 is [[5]]
```

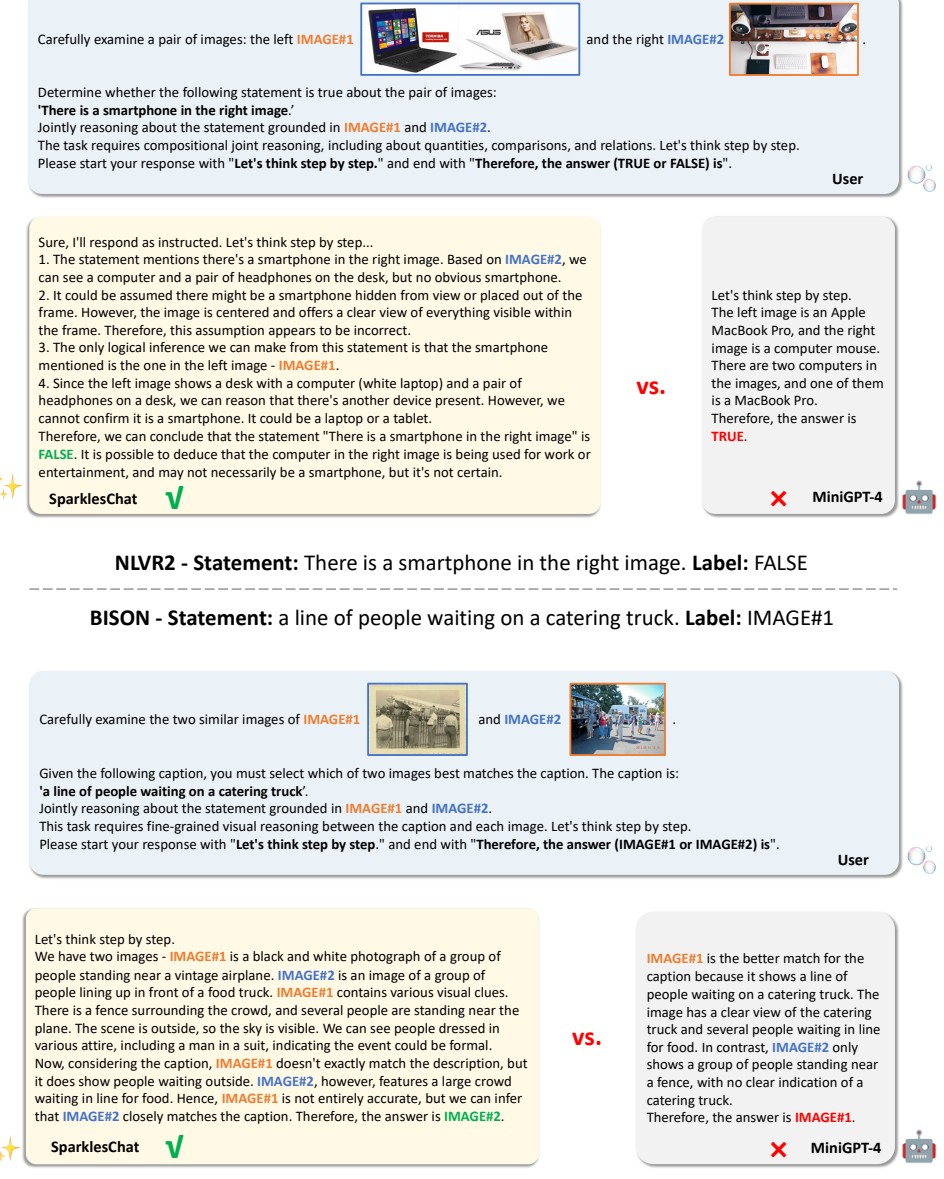

Figure 5: Comparison between SparklesChat and MiniGPT-4 on examples of NLVR2 and BISON.

2022), we combined the reasoning extraction and answer extraction stages into a single prompt: *"Please start your response with 'Let's think step by step.' and end with 'Therefore, the answer is'"*. The full evaluation prompt templates to evaluate NLVR2 and BISON datasets are presented in Table 6. We regenerated the response if the model failed to follow the instructions to output responses in the specified format. This approach ensures an unambiguous response and allows us to extract a potential answer from the text following the last occurrence of *"Therefore"*.

## F DEMONSTRATIONS AND APPLICATIONS

We conducted qualitative demonstrations to showcase the model's wide applications in free-form scenarios by asking questions such as: "Create a story that takes place in ⭐ for the characters depicted in ⭐.", "Imagine a dialogue between Harry Potter and ⭐ that takes place in the scene of ⭐.", "Create a song where the scene twists from ⭐ to ⭐.", "Create a title for this song that takes inspiration from ⭐.". Examples in Figure 6, Figure 7, and Figure 8 demonstrate two or three-turn dialogues,

Table 6: Prompt formats to evaluate NLVR2 and BISON datasets.

**NLVR2**
Carefully examine a pair of images: the left IMAGE#1<ImageHere></Img> and the right IMAGE#2<ImageHere></Img>.
Determine whether the following statement is true about the pair of images:
'{Statement}'
Jointly reasoning about the statement grounded in IMAGE#1 and IMAGE#2.
The task requires compositional joint reasoning, including quantities, comparisons, and relations.
Let's think step by step.
Please start your response with "Let's think step by step." and end with "Therefore, the answer (TRUE or FALSE) is".

———————————————————————————————————

**BISON**
Carefully examine the two similar images of IMAGE#1<ImageHere></Img> and IMAGE#2<ImageHere></Img>.
Given the following caption, you must select which of two images best matches the caption.
The caption is: '{Caption}'.
This task requires fine-grained visual reasoning between the caption and each image. Let's think step by step.
Please start your response with "Let's think step by step." and end with "Therefore, the answer (IMAGE#1 or IMAGE#2) is".

with each turn involving one or two images. SparklesChat is capable of distinguishing different images interleaving with text within and across dialogue turns. To analyze SparklesChat's ability to generalize in scenarios involving a larger number of images, we consolidated several questions from a dialogue into a single query. This was done to generate responses for queries involving four or five images, as shown in Figure 9 and Figure 10, where the model effectively distinguishes between the images and adheres to complex queries. In the case of Figure 10, where three questions involving five images are concatenated into one query, the model tends to ignore the final question and only responds to the first two. We believe this limitation arises from the absence of similar patterns in the training data. A potential solution could involve incorporating multiple turns into each training dialogue to enhance the model's ability to handle such complex scenarios.

## G  JUDGING WITH DIFFERENT VERSIONS OF GPT MODELS

As of July 2023, while it is widely recognized that employing `gpt-4` as a judge model outperforms alternatives such as `gpt-3.5-turbo`, the cost of using `gpt-4` is significantly higher. Therefore, we also provide scores generated by `gpt-3.5-turbo` as a reader reference, although we strongly recommend utilizing `gpt-4` or more advanced future models as reliable judges. The version of `gpt-4-0701` refers to API version `2023-07-01-preview` for the GPT-4 model. We adopt the latest version `gpt-4-0701` as our default judge model.

We evaluate GPT-4, MiniGPT-4, and SparklesChat using SparklesEval, leveraging three versions of judge models, as presented in Table 7. Both MiniGPT-4 and SparklesChat generate responses based on the question and accompanying visual image. At the same time, GPT-4 is a reference LLM that only uses textual information, including the question, the ground-truth bounding boxes, and captions. From the table, we observe that the more advanced judge models - `gpt-4-0613` and `gpt-4-0701` - provide higher scores compared to the older `gpt-3.5-turbo-0613` when assessing both GPT-4 and OurModel (approximately nine versus eight). However, these advanced judge models yield considerably lower scores for MiniGPT-4 (about three versus five). GPT-4 achieves the highest score of 9.26 out of 10 when evaluated by the default `gpt-4-0701` mainly due to its use of detailed ground-truth annotations. Nevertheless, it's worth noting LLM judge models may display a self-enhancement bias, favoring the responses they generate (Zheng et al., 2023). In contrast, MiniGPT-4 performs behind with a score of just 3.91. SparklesChat achieves a score of 8.56 - about 92% of the GPT-4 score - demonstrating SparklesChat's efficacy in generating responses that are not only relevant and complete but also exhibit cross-image and cross-turn coherence.

Create a story that takes place in 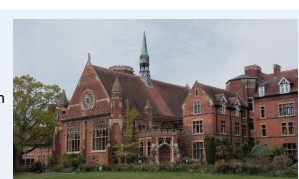 for the characters depicted in 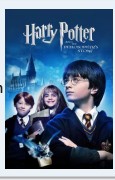 .

**User**

Once upon a time, in the heart of the city, there stood a grand, red brick building - the Hogwarts School of Witchcraft and Wizardry. It was a place where students from all over the world came to learn the magical arts and explore their hidden talents. Harry Potter, the protagonist of the story, had always dreamed of attending Hogwarts. He was an orphan, and his mother's death had left him feeling isolated and alone. But, he was determined to fulfill his mother's dream of becoming a wizard.

The day Harry arrived at Hogwarts, he was struck by the grandeur of the building. The tall, steepled clock tower stood out against the bright blue sky, and the red brick walls seemed to glow in the sunlight. The students were all dressed in their robes, and Harry couldn't help but feel a sense of awe and wonder.

As he made his way through the castle, Harry was greeted by the warm, friendly faces of his new classmates. He was struck by the diversity of the students, each with their own unique talents and personalities. Some were friendly, while others seemed aloof, but Harry was determined to make the most of his time at Hogwarts.

In his first year at Hogwarts, Harry was sorted into Gryffindor House, where he met his best friends Ron Weasley and Hermione Granger. Together, they faced many challenges and adventures, from the dangerous Dementors to the infamous Heir of Salazar Slytherin. Through it all, Harry learned the true meaning of friendship, bravery, and the power of love.

Year after year, Harry continued his magical education at Hogwarts. He learned how to cast spells, brew potions, and explore the mysteries of the wizarding world. He also discovered his own magical talents, such as his ability to speak Parseltongue and his connection to Voldemort's past.

As Harry's time at Hogwarts drew to a close, he realized that he had truly found his home. He had friends, a sense of belonging, and a deep understanding of the magical world. And, most importantly, he had learned the true value of love, friendship, and bravery. In the end, Harry's journey at Hogwarts had transformed him from a shy orphan into a powerful wizard, ready to face the challenges of the future.

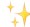 **SparklesChat**

Imagine a dialogue between Harry Potter and 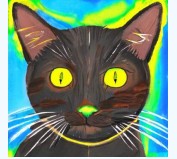 that takes place in the scene of 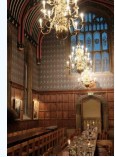 .

**User**

Harry Potter, a young wizard, stands in front of the grand dining hall of Hogwarts School of Witchcraft and Wizardry. The room is decorated with wooden tables, chairs, and a large chandelier hanging from the ceiling. The walls are adorned with tapestries and paintings depicting various magical creatures and scenes from Hogwarts' history. Harry's eyes are fixed on the black cat sitting on the table in front of him.

Harry: "Hello there, Mr. Whiskers."

Mr. Whiskers: "Hello, Harry. How are you doing today?"

Harry: "I'm doing well, thank you. I'm just admiring the beauty of this dining hall. It's quite a sight, isn't it?"

Mr. Whiskers: "Indeed, it is. Hogwarts has always been a place of grandeur and magic. I'm glad you appreciate it."

Harry: "I've been looking forward to this day for a long time. It's my first day at Hogwarts."

Mr. Whiskers: "Ah, I remember my first day at Hogwarts. It was quite overwhelming, but also thrilling. I'm sure you'll have a wonderful time here."

Harry: "I hope so. I've heard so much about Hogwarts and the magical world. I can't wait to explore it all."

Mr. Whiskers: "Well, you're in the right place. Hogwarts is full of magic, adventure, and wonder. I'm sure you'll find your place here."

Harry: "Thank you, Mr. Whiskers. I'm looking forward to discovering it all."

**SparklesChat**

Figure 6: Demonstration of SparklesChat to create a story and a dialogue that connects places and characters.

Create a song where the scene twists from [image] to [image].

**User**

The song you've imagined is a beautiful blend of imagination and reality. The opening scene depicts a young girl in a red dress standing on the edge of a cliff, looking out at a bustling city below. She's holding onto a long rope, which seems to be tied to a wooden structure on the edge of the cliff. The girl looks peaceful and serene, as if she's contemplating something profound.

As the song progresses, the scene shifts to a vibrant and colorful world of balloons. The girl is now in the middle of a bustling city, surrounded by colorful balloons of all shapes and sizes. The balloons seem to be floating in the air, creating a sense of playfulness and joy.

The lyrics of the song could be something like this:

Verse 1:
She's standing on the edge of a cliff, looking out at the city below
She's holding onto a long rope, tied to a wooden structure on the edge of the cliff
She's lost in thought, as if she's contemplating something profound

Chorus:
She's floating in a sea of balloons, surrounded by joy and playfulness
She's laughing and smiling, as if she's found a new sense of freedom
She's dancing with the wind, as if she's found a new way to fly

Verse 2:
She's in the middle of the city, surrounded by colorful balloons of all shapes and sizes
She's bouncing and leaping, as if she's found a new way to live
She's laughing and smiling, as if she's found a new way to be free

Chorus:
She's floating in a sea of balloons, surrounded by joy and playfulness
She's laughing and smiling, as if she's found a new sense of freedom
She's dancing with the wind, as if she's found a new way to fly

Bridge:
She's found a new way to soar, a new way to fly
She's found a new way to live, a new way to be free
She's found a new way to dream, a new way to be

Chorus:
She's floating in a sea of balloons, surrounded by joy and playfulness
She's laughing and smiling, as if she's found a new sense of freedom
She's dancing with the wind, as if she's found a new way to fly.

**SparklesChat**

Create a title for this song that takes inspiration from [image] and tell how they are connected

**User**

The song could be titled "Golden Stars" as a nod to the image of the golden stars in the sky. The image of the stars in the sky could be interpreted as a metaphor for the dreams and aspirations of the protagonist in the song. The stars in the sky are bright, shining, and inspiring, much like the protagonist's dreams and aspirations in the song. The song's chorus "She's floating in a sea of balloons, surrounded by joy and playfulness" could be interpreted as a metaphor for the protagonist's journey towards her dreams and aspirations, much like the way the balloons in the image are floating and bouncing in the air. The image of the stars in the sky and the balloons in the song both represent a sense of hope, aspiration, and dreams.

**SparklesChat**

Figure 7: Demonstration of SparklesChat to compose a song containing two scenes and generate a song title inspired by another image.

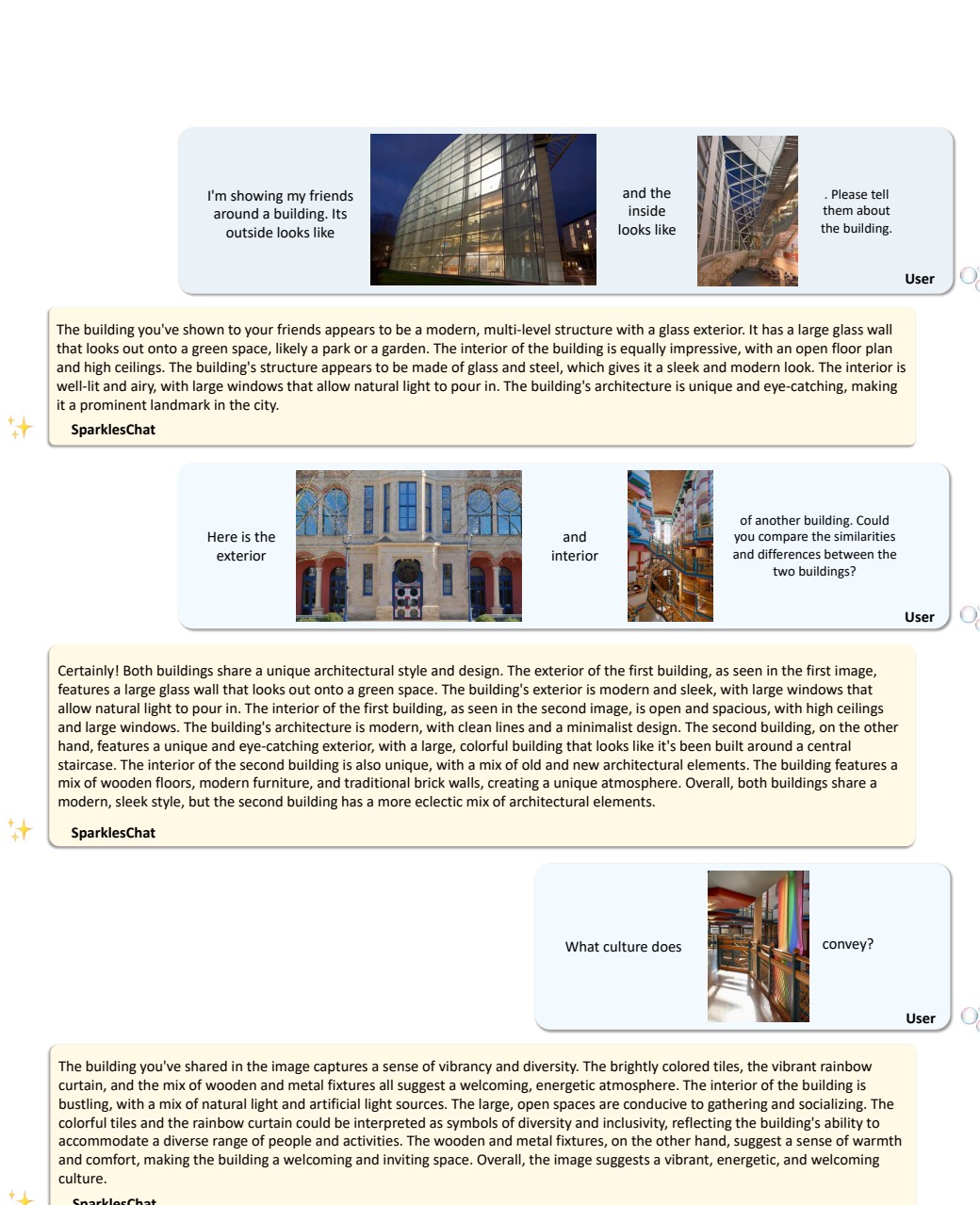

Figure 8: Demonstration of SparklesChat to describe and reason about different groups of images.

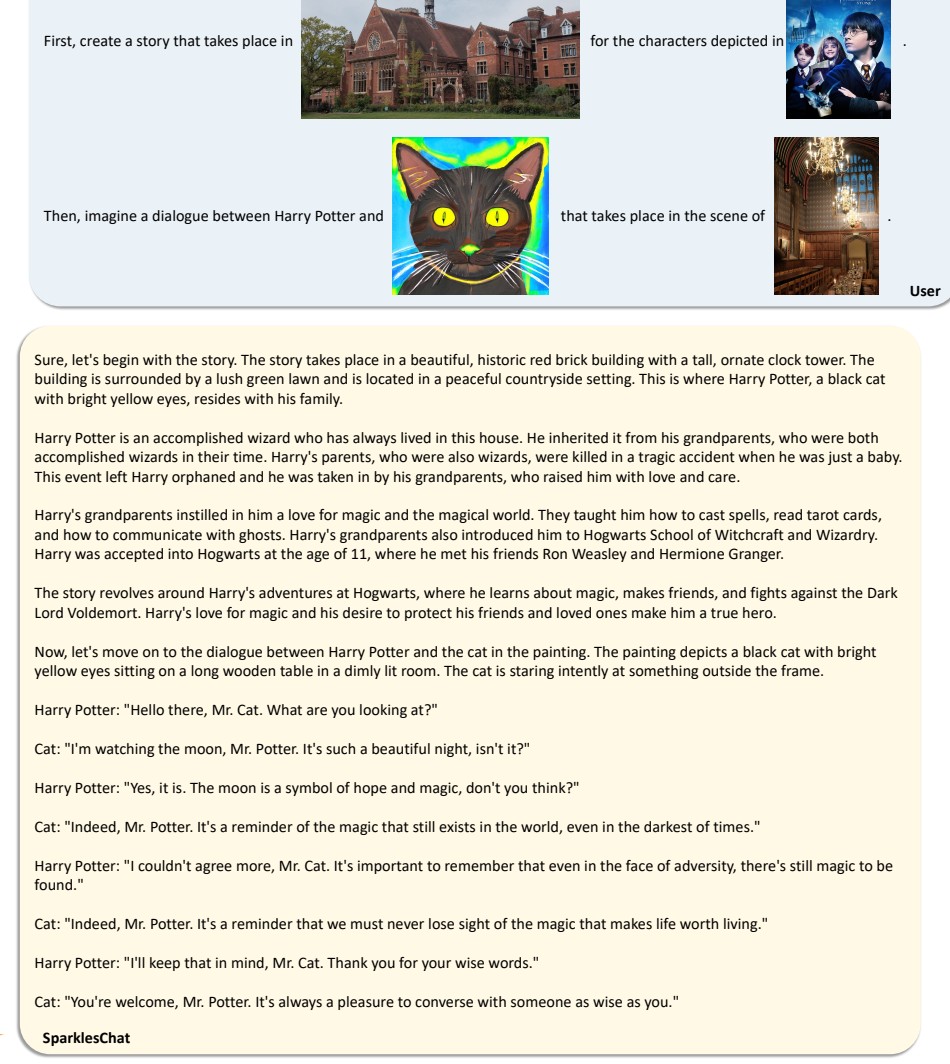

Figure 9: Demonstration of SparklesChat to respond to a question with four images to create a story and a dialogue that connects places and characters.

## H CHARACTERISTICS AND VERB-NOUN DISTRIBUTION ANALYSIS

The characteristics of SparklesDialogueCC are shown in Figure 11. For verb-noun distribution, we follow Self-instruct (Wang et al., 2022) to extract the verb closest to the root and its first direct noun object and plot the top 20 most common root verbs and their top 4 direct noun objects. We use the Berkeley Neural Parser[6] (Kitaev & Klein, 2018) to parse user messages. We mainly focus on the last sentence of each message because it usually contains the question. If we can't extract the verb-noun pair from it, we look at the first sentence instead. For SparklesDialogueVG and SparklesDialogueCC, we visualize the verb-noun distributions regarding different numbers of images in each turn in Figure 12 and Figure 13 respectively.

---

[6] https://parser.kitaev.io/

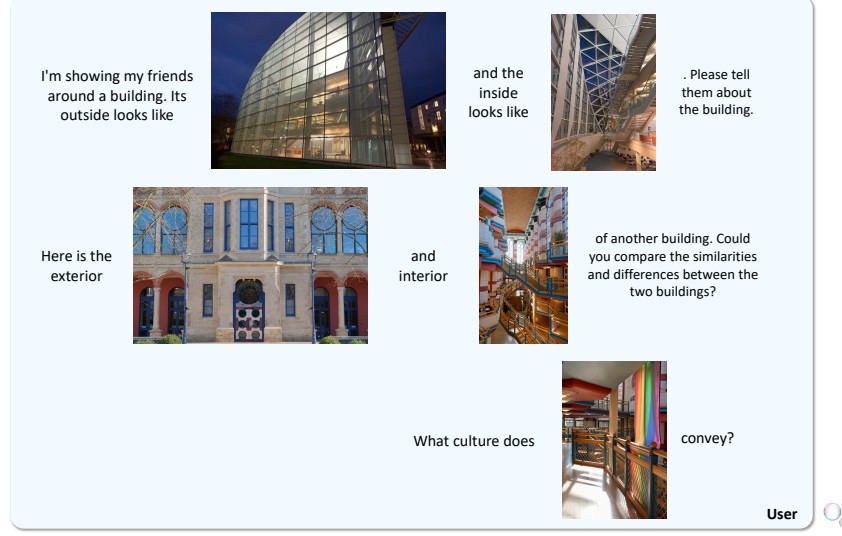

Figure 10: Demonstration of SparklesChat to respond to a question with five images to describe and reason about different groups of images.

Table 7: Evaluation results on SparklesEval with different judge models.

| Model | Judge Model Version | Score | Turn one | | | | Turn two | | | |
|---|---|---|---|---|---|---|---|---|---|---|
| | | | A1 | C1 | C2 | C3 | A2 | C1 | C2 | C3 |
| GPT-4 | gpt-3.5-turbo-0613 | 8.48 | 8.61 | 8.69 | 8.18 | 8.95 | 8.35 | 8.27 | 8.08 | 8.68 |
| | gpt-4-0613 | 9.51 | 9.50 | 9.53 | 9.37 | 9.60 | 9.53 | 9.49 | 9.46 | 9.64 |
| | **gpt-4-0701** | 9.26 | 9.26 | 9.23 | 9.18 | 9.38 | 9.26 | 9.25 | 9.15 | 9.38 |
| MiniGPT-4 | gpt-3.5-turbo-0613 | 5.51 | 5.46 | 6.11 | 4.78 | 5.48 | 5.55 | 5.92 | 5.23 | 5.51 |
| | gpt-4-0613 | 3.31 | 3.11 | 3.12 | 3.09 | 3.10 | 3.51 | 3.57 | 3.40 | 3.56 |
| | **gpt-4-0701** | 3.91 | 3.55 | 3.67 | 3.53 | 3.44 | 4.28 | 4.38 | 4.21 | 4.23 |
| SparklesChat | gpt-3.5-turbo-0613 | 8.37 | 8.51 | 8.59 | 8.04 | 8.89 | 8.24 | 8.16 | 7.92 | 8.64 |
| | gpt-4-0613 | 8.82 | 8.75 | 8.78 | 8.59 | 8.89 | 8.88 | 8.89 | 8.79 | 8.97 |
| | **gpt-4-0701** | 8.56 | 8.76 | 8.81 | 8.67 | 8.81 | 8.35 | 8.37 | 8.28 | 8.41 |

# I LIMITATIONS AND FUTURE WORKS

We discuss some limitations of this work to inspire future research in this field. First, SparklesChat shares common drawbacks with large language models, such as being out-of-date in its knowledge, sometimes providing inaccurate information, and having limited context length and inference speed (Openai, 2023). Potential solutions may include regular updates to the model's knowledge base and fine-tuning with more reliable data sources. Second, SparklesChat inherits weaknesses from vision models, such as inaccurate object recognition, people/places identification, or visual relationships reasoning (Li et al., 2023c). This calls for a more powerful visual perception model, and training on more well-aligned image-text datasets. Third, SparklesChat occasionally encounters

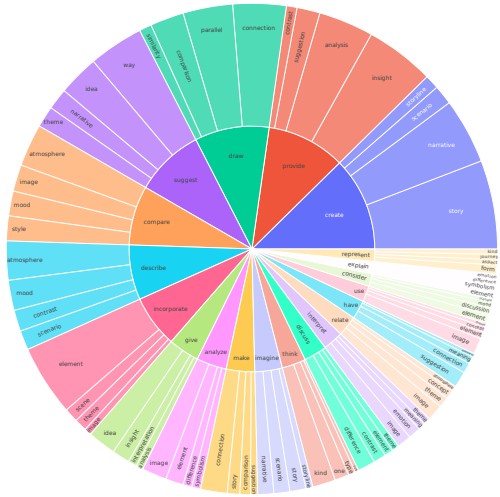

(a) Root verb-noun pairs in user messages.

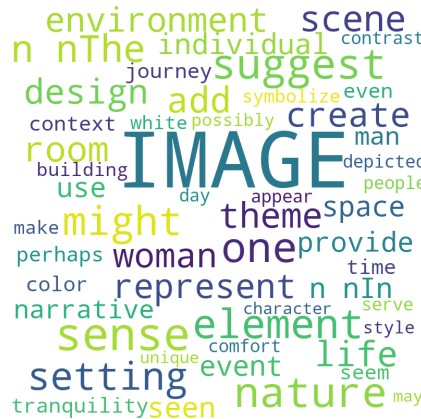

(b) Word cloud of assistant messages.

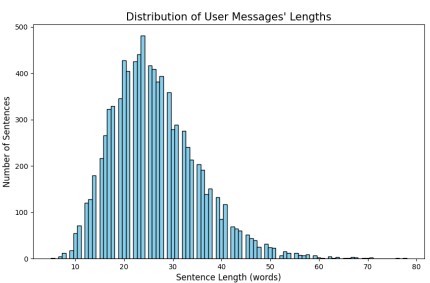

(c) Distribution of word lengths in user messages with an average of 26.3 words.

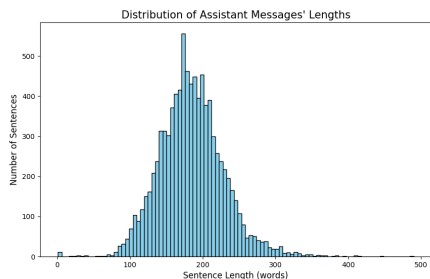

(d) Distribution of word lengths in assistant messages with an average of 184.6 words.

Figure 11: Characteristics of SparklesDialogueCC.

difficulties maintaining multi-image and multi-turn consistency. Specifically, the model may lose the context of prior images after several dialogue turns or mix up the contents of different images. Potential solutions involve advanced model designs in position encoding and attention mechanisms to enhance the model's consistency in recalling historical images and dialogues. Fourth, Sparkles-Dialogue primarily concentrates on natural images, which limits its versatility in handling text-rich images such as charts, tables, and receipts, as well as domain-specific images such as medical scans, math illustrations, and satellite photos. Moreover, the dialogues in SparklesDialogue do not cover all possible user scenarios. Therefore, broadening the dataset to cover more diverse image types and user cases is a direction for future work. Fifth, the reliability of SparklesEval is tied to the capabilities of current GPT models. This limitation can be mitigated by incorporating more robust judge models and the assistance of human evaluators. Lastly, further safety considerations are needed to mitigate potential misuse of the model. Future works addressing these issues should make for a more reliable and robust system.

## J  GPT-ASSISTED SPARKLESDIALOGUE GENERATION

### J.1  SINGLE DIALOGUE GENERATION FOR SPARKLESDIALOGUEVG

For SparklesDialogueVG, we generate one two-turn dialogue at a time, with the first turn incorporating two or three images. We derive the demonstration dialogues from SparklesDialogueCC to encourage diversity. However, to minimize redundancy, we retain only those dialogues with unique verb-noun combinations in the user questions. This results in pools of 661 and 441 demonstration dialogues

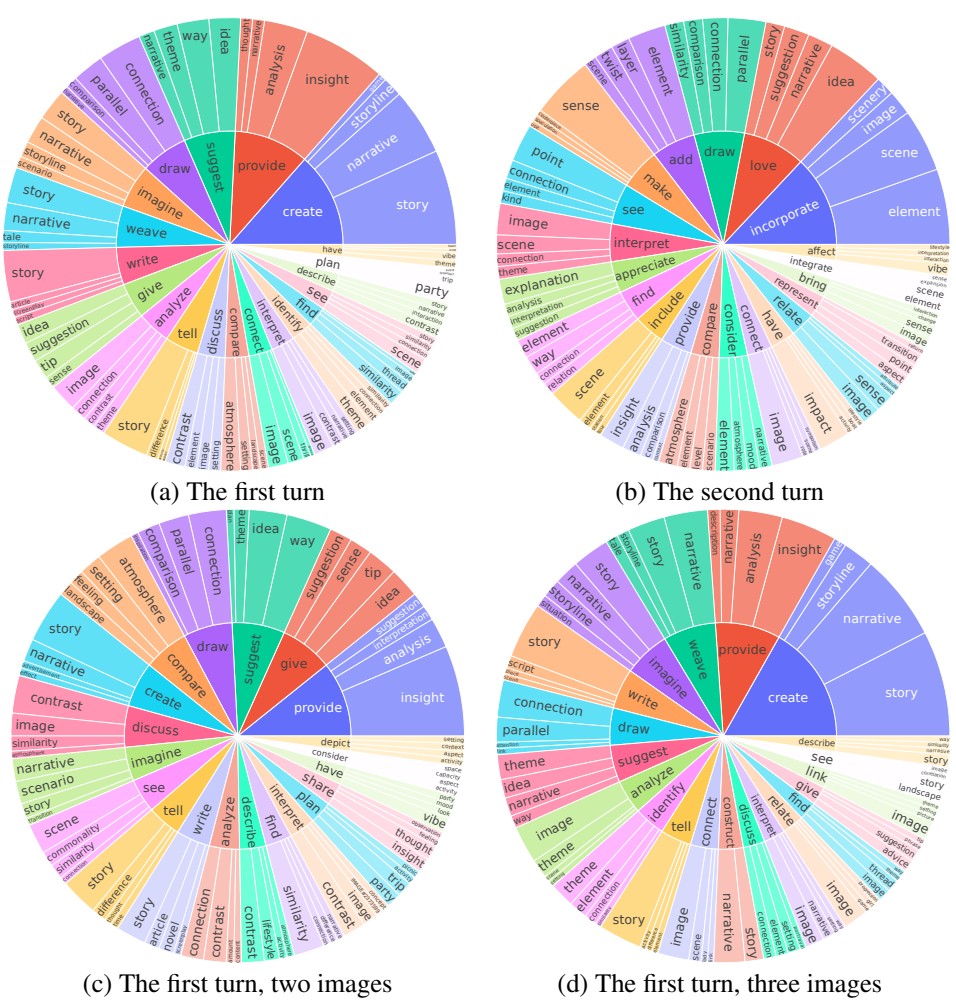

(a) The first turn

(b) The second turn

(c) The first turn, two images

(d) The first turn, three images

Figure 12: Root verb-noun distributions of SparklesDialogueVG.

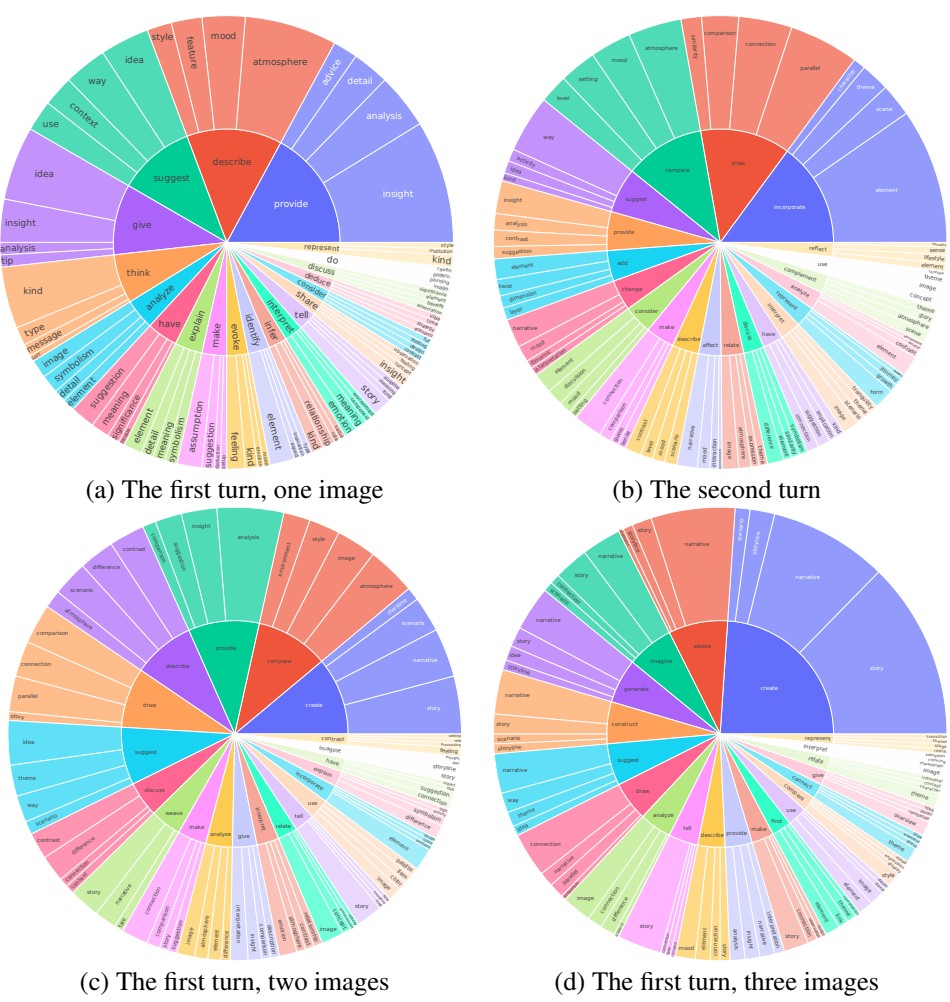

(a) The first turn, one image

(b) The second turn

(c) The first turn, two images

(d) The first turn, three images

Figure 13: Root verb-noun distributions of SparklesDialogueCC.

for conversations incorporating two or three images in the first turn, respectively. We pull from an expansive collection of roughly 100,000 image-text pairs for this dataset. We randomly select four candidates each time, and they are not reused by excluding them from future selections.

We first present our designed **prompt** for GPT-assisted Single Dialogue Generation to generate SparklesDialogueVG in Table 8. Then, we show a case of the `Dialogue Demonstration` and `Candidate Image Descriptions` to construct the prompt. Finally, we show the corresponding **generated dialogue** using the example prompt.

**Example of dialogue demonstration.** We visualize the images corresponding to image IDs in the dialogues in Figure 14 for reference, while these visual images were not sent to GPT-4 for data generation. Note that we abbreviate the message content of the assistant in the second turn as "..." to save space, considering that the previous message contents have provided enough demonstrations.

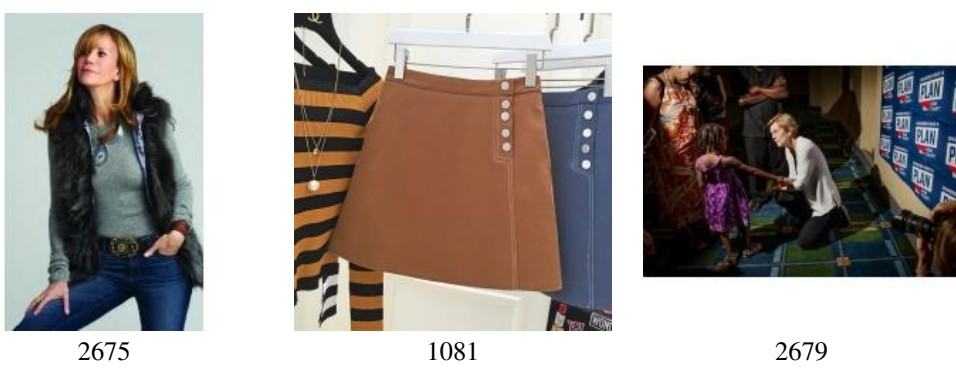

2675              1081              2679

Figure 14: Reference images corresponding to the image IDs in the demonstration dialogues in section J.1. These images were not sent to GPT-4 for data generation.

```json
[
[
    {'role': 'user', 'image_ids': [2675, 1081], 'content': 'How would you
        connect the woman in IMAGE#2675 with the clothing items in IMAGE
        #1081?'},
    {'role': 'assistant', 'content': 'In IMAGE#2675, we see a woman
        dressed in fashionable attire consisting of a grey turtleneck
        sweater, blue jeans, a black leather belt, a black leather jacket
         with fur detailing and black ankle boots. Her overall outfit
        looks stylish, suggesting she has good fashion sense.\\\\n\\\\
        nIMAGE#1081 represents a group of clothing items hanging on a
        rack. It shows an outfit comprising a striped shirt, a brown
        skirt, and a pair of black shoes. The clothes and shoes have a
        sophisticated touch to them.\\\\n\\\\nConnecting these images,
        one could infer that the clothing items could belong to the same
        woman. Maybe she's someone who likes keeping her wardrobe
        organized or could be in the fashion industry focusing on outfit
        planning. Her style in IMAGE#2675 is modern and chic, while the
        ensemble in IMAGE#1081 is equally trendy, with classic
        combinations of stripes and brown, which many consider a style
        staple, and the elegant black shoes. Therefore, we could imagine
        that the outfit from IMAGE#1081 may be another stylish option
        that she might opt for. The clothing items could be a part of her
         personal style or they might signal her work attire, given the
        formal vibe the striped shirt and brown skirt ensemble radiate.'}
        ,
    {'role': 'user', 'image_ids': [2679], 'content': 'Looking at IMAGE#26
        79, how would you weave it into the narrative of this stylish
        woman from IMAGE#2675 and the clothes in IMAGE#1081?'},
    {'role': 'assistant', 'content': '...'}
]
```

Table 8: Prompt for GPT-assisted single dialogue generation.

---

**System: You are a helpful assistant.**
Users will interact with a conversational assistant that has advanced capabilities of understanding, analyzing, and reasoning about images. This includes discussing a variety of real-world concepts, objects, and entities, generating a range of text materials, seeking advice, guidance, or assistance, and much more.

Below is an illustrative dialogue presented in a JSON format. The dialogue represents a meaningful conversation between a "user" and the "assistant" regarding multiple images. Each "user" message contains an "image_ids" field recording the IDs of newly selected images. The images are referred to in the "content" field as IMAGE#image_id.

```json
{Dialogue Demonstration}
```

Please note that the user contents in the JSON above may be a counterexample that reveals the content of images and can be answered without looking at the images. Please make sure not to reveal the content of the images or describe the images in the user messages in the conversation that follows.

Please note that the specific "image_ids" and "content" in the JSON above are for illustrative purposes only. The actual candidate images are shown below delimited by triple quotes, each accompanied by an image ID and a caption. Avoid using phrases similar to 'caption' and 'description' in your dialogue as if the user and the assistant have visual capabilities.

```json
{Candidate Image Descriptions}
```

Each dialogue consists of four messages:
1. A user examines all candidate images, selects {Number of Images} highly relevant images, and sends a reasonable and creative message to the assistant.
2. Once the images are provided, the assistant thoroughly perceives and comprehends them, responding with highly helpful and exceptionally detailed answers that provide comprehensive reasoning regarding the visual content of the images.
3. Considering the past dialogue, the user chooses other candidate images for further inquiry. The user should refer to both the newly selected images and those mentioned earlier in the same dialogue.
4. The assistant provides a highly helpful and exceptionally detailed answer providing comprehensive reasoning regarding the visual content of the images.

The following is a dialogue between the user and the assistant, adhering to the given JSON format. Make sure to formulate accurate and diverse "content" that does not follow the illustrative dialogues. And remember to develop the last "content" even though it is shown as "..." in the JSON format provided above.

---

Dialogue Demonstration

```
[[{'role': 'user', 'image_ids': I^{1,1}, 'content': X_q^{1,1}},
  {'role': 'assistant', 'content': X_a^{1,1}},
  {'role': 'user', 'image_ids': I^{1,2}, 'content': X_q^{1,2}},
  {'role': 'assistant', 'content': '...'}]]
```

Candidate Image Descriptions

```
[{'image_id': J^1, 'caption': C^1},
 ......
  {'image_id': J^4, 'caption': C^4}]
```

Number of Images is "two" or "three".

```
9   ]
10  ```
```

**Example of candidate image descriptions.** The image IDs in the dialogues refer to the images in Figure 15.

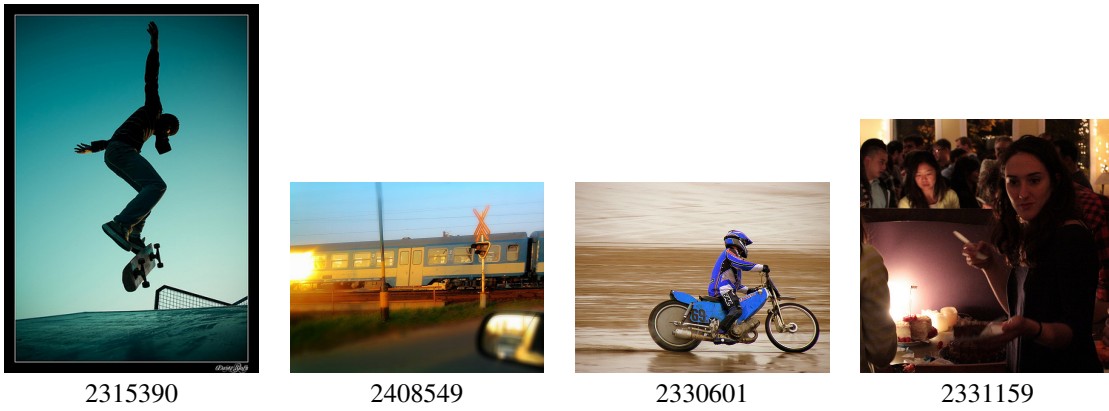

| 2315390 | 2408549 | 2330601 | 2331159 |

Figure 15: Candidate images corresponding to the image IDs in the dialogues generation process in section J.1. These images were not sent to GPT-4 for data generation.

```
1   ```json
2   [
3       {'image_id': 2315390, 'caption': 'The image captures a dynamic scene
            at a skate park, specifically a boy mid-air, performing a trick
            on his skateboard. The boy appears to be a bit on the dark side,
            silhouetted against the lighter sky behind him. He's adorned in
            casual skate attire: a sweater, blue jeans, and athletic shoes.
            His arms are raised in the air, which adds to the intensity of
            the trick he's executing. Interestingly, he is not wearing a
            helmet.\\nThe skateboard itself is flipped, and it appears to be
            on its side. This unusual positioning gives the impression that
            the skater is performing an intricate and complex trick. The
            skateboard has multiple wheels, described as black, and is
            noticeably detailed in the image.\\nThere's another person
            present in the scene, presumably a photographer or a spectator,
            positioned toward the right corner of the frame. However, this
            individual is located quite close to the edge, suggesting that
            they are not the main focus of the photograph.\\nThe skate park
            consists of a grey concrete ramp that the boy is using for his
            tricks. It stretches across the majority of the bottom part of
            the picture, a hard, flat contrast to the dynamic motion taking
            place above it. There's also a metal gate visible in the scene,
            possibly part of the boundary or safety measures at the park.\\
            nThe backdrop is a vibrant blue sky with clusters of white clouds
             scattered across it. It seems to be a bright, clear day, perfect
             for outdoor activities. Lastly, a safety net in the distance
            lends an additional element of safety to the environment.\\
            nOverall, the photograph encapsulates an exhilarating moment of
            skill, action, and athleticism at a bustling skate park, set
            against a serene, blue-skied day.'},
4       {'image_id': 2408549, 'caption': "This image captures a dynamic scene
             of a large blue train moving rapidly on railroad tracks. The
            train's hue stands in beautiful contrast with the clear, blue sky
             overhead. The train is quite long, stretching almost the entire
            width of the image, and it appears to be well maintained, with
            grey stripes highlighting its design. The train's lower half is
            primarily filled with windows and double doors. Three windows are
             clearly visible, each reflecting the bright sunlight.\\nWithin
```

the train, passengers can be seen through the windows. Notably,
one person dressed in a white t-shirt is looking out of the
window, taking in the scenery or perhaps observing the vehicle
whose side mirror is captured in the frame. The double doors, one
 on the left and the other on the right, stand out on the body of
 the train. Each door has a number '2' inscribed on it.\\
nInterestingly, in the right section of the image, the side
mirror of a car is in the frame, reflecting a blurry image of
another vehicle, further contributing to the sense of movement in
 this scene.\\nThe foreground of the image is filled with a wide
expanse of green grass that contrasts nicely with the railroad
tracks and a nearby road. To the right of the train, there's a
tall pole that rises high into the image, likely used for
mounting signs. In this case, the pole hosts a railroad crossing
sign with lights and a large X on top. There is also a triangular
 sign with three lights underneath the X sign, providing
important safety information for approaching vehicles.\\nBehind
the pole, a red metal barrier is barely visible. It appears to be
 part of the infrastructure that surrounds the tracks. With the
beautiful sunny sky overhead, this picture seems to represent a
typical day with normal hustle and bustle at this railroad
crossing. The sunlight reflecting off the train windows adds a
stunning glow to the scene.\\nDespite the fast motion of the
train, details such as the wheels and even the driver's side view
 mirror are captured in the image, emphasizing the skill of the
photographer in capturing this dynamic and detailed snapshot of a
 moment in time."},

5        {'image_id': 2330601, 'caption': 'This image depicts an exciting
         scene of a man dressed in a blue and black racing suit, riding a
         dirt bike on a muddy track. The man is prominently positioned in
         the image, seeming to occupy a considerable portion of it from
         left to right. His blue helmet, matching his attire, is clearly
         visible.\\nThe motorcycle he's racing is intricately detailed.
         Its prominent front and back black wheels kick up wet mud as they
          tear through the track, while the metallic shimmer of the
         exhaust and the sturdy grey frame suggest its rugged durability.
         A number, black in color, stands out on the side of the bike, and
          there's a patch of blue at the bike's back that contributes to
         the cohesive color scheme.\\nThe rider's attire stands out as
         well. Apart from the matching helmet, he's wearing a blue and
         orange shirt, black pants, and blue and yellow shoes. A black
         visor on his helmet and black gloves further accessorize his
         ensemble. His coat, in shades of blue and grey, fits snugly,
         outlining his physique.\\nThe scene around the bike is as dynamic
          as the rider. The track underneath is a dark brown, most likely
         a mix of dirt and water, suggestive of recent rain or the
         challenging conditions of a dirt bike race. Patches of water and
         water spots can be seen at various locations, indicating the
         wetness of the track and the splashing caused by the bike
         speeding through.\\nMoreover, there's an evident sense of motion
         in the image with water splashing up from the bike and wet sand
         scattering in its wake. The ground can be seen in patches,
         displaying its dark brown color. Amidst all this action, the bike
          stands as a striking subject in the image, catching the eye with
          its blue frame and detailings, while the rider, dressed in
         coordinating colors, charges forward.\\nAll of these elements
         combined create an image that is full of life and action,
         capturing a thrilling moment of a dirt bike race in progress.'},

6        {'image_id': 2331159, 'caption': 'The image is lively, filled with
         people gathered possibly for a party or a social event. In the
         center of the image, a woman dressed elegantly draws attention.
         She stands prominently, making a distinct statement with her long
         , dark hair. Her face, sharply defined, features a noticeably
         distinct nose. She is holding a white plastic spoon in one hand,
         which also showcases a black wristwatch. As she raises the spoon,

```
            it's clear that she is indulging in a delicious treat, a piece
            of cake resting on a small plate.\\nThis cake is a stunning
            creation, white with red frosting. It's adorned with a delightful
             mix of red strawberries and an array of white candles. On the
            cake, there seems to be a flag as well, perhaps signifying a
            special occasion. An unused serving knife rests in the cake, and
            it appears that the woman has just served herself a piece.\\
            nAround her, numerous other faces peer out, all engaged in their
            individual conversations. Most of them appear to be men, some
            notable for their long hair and glasses. There's an interesting
            mix of attire in the scene, from yellow and white-striped shirts
            to red and black plaid ones. One Asian man to the right seems to
            be focused on the woman with the cake, adding to the collective
            sense of attention directed towards her.\\nIn the background,
            several intriguing details pop out. For instance, the twinkling
            lights to the right catch the eye, likely part of the party's
            decoration. Additionally, there is a mirror behind the woman,
            reflecting the attendees and amplifying the sense of a bustling
            crowd. A lamp stands beside a wall, casting a warm glow, while
            the silhouettes of patrons in the dark restaurant form an
            atmospheric backdrop. A window lets in some additional light,
            illuminating a vacant chair.\\nThe overall atmosphere conveys the
             joyous, friendly nature of the gathering. You can almost hear
            the buzz of conversation and feel the warmth of shared laughter.
            It's clear that this is an occasion of happiness and togetherness
            .'}
7 ]
8 ```
```

**Dialogue example from SparklesDialogueVG.** The generated dialogue is visualized in Figure 16. The raw text is shown as follows. The image IDs in the dialogue refer to the images in Figure 15.

```
1 ```json
2 [
3   [
4       {'role': 'user', 'image_ids': [2331159, 2330601], 'content': 'Can
            you link the celebration occurring in IMAGE#2331159 and the dirt
             bike race in IMAGE#2330601?'},
5       {'role': 'assistant', 'content': 'Certainly, in IMAGE#2331159, we
            see a lively gathering with a woman at the centre of the
            attention. She is seen grasping a slice of intricately decorated
             cake from a vibrant party of sorts. Her jovial demeanor and
            elegant apparel suggest she may be the guest of honor at this
            event. The ambiance in the background features warm lighting,
            laughter, conversation, and a sense of togetherness and
            happiness. This scene could possibly be a celebration of a
            personal achievement or milestone.\\n\\nOn the other hand, in
            IMAGE#2330601, a different form of excitement and thrill is
            captured. This showcases a man in a blue and black racing suit,
            performing at a dirt bike race, implying a high level of
            commitment, skill, and passion. He is decisively navigating the
            muddy track, indicating a race in progress.\\n\\nLinking these
            images, one possible narrative could be that the woman in the
            first image might be celebrating a successful day at the dirt
            bike race. The man in the second image who is skilfully
            navigating the muddy race track could be related to her in some
            way. Perhaps he is her son, husband, or even a team member, and
            they are celebrating his win or a significant accomplishment in
            his racing career at the party. The sense of joy and exhilarity
            is a common thread that connects these two images.'},
6       {'role': 'user', 'image_ids': [2408549], 'content': 'How would you
            incorporate the train scene from IMAGE#2408549 with the earlier
            party and racing narratives?'},
```

Can you link the celebration occurring in **IMAGE#2331159** 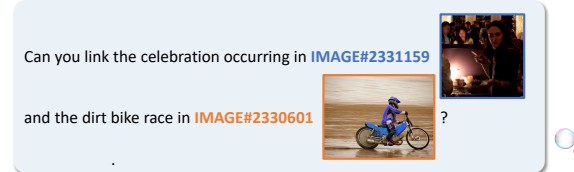

and the dirt bike race in **IMAGE#2330601** 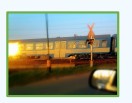 ?
.

Certainly, in **IMAGE#2331159**, we see a lively gathering with a woman at the centre of the attention. She is seen grasping a slice of intricately decorated cake from a vibrant party of sorts. Her jovial demeanor and elegant apparel suggest she may be the guest of honor at this event. The ambiance in the background features warm lighting, laughter, conversation, and a sense of togetherness and happiness. This scene could possibly be a celebration of a personal achievement or milestone.

On the other hand, in **IMAGE#2330601**, a different form of excitement and thrill is captured. This showcases a man in a blue and black racing suit, performing at a dirt bike race, implying a high level of commitment, skill, and passion. He is decisively navigating the muddy track, indicating a race in progress.

Linking these images, one possible narrative could be that the woman in the first image might be celebrating a successful day at the dirt bike race. The man in the second image who is skilfully navigating the muddy race track could be related to her in some way. Perhaps he is her son, husband, or even a team member, and they are celebrating his win or a significant accomplishment in his racing career at the party. The sense of joy and exhilarity is a common thread that connects these two images.

How would you incorporate the train scene from **IMAGE#2408549** with the earlier party and racing narratives?

In **IMAGE#2408549**, we witness the rush of a large, striking blue train speeding through a railroad crossing. A passenger, perhaps contemplating the journey, watches the world outside through a window. This train scene can signify a journey or transition.

Bringing these images together into a coherent story, it might be speculated that after the celebration in **IMAGE#2331159** for the successful dirt bike race from **IMAGE#2330601**, the next chapter of their journey is starting. It could be speculated that the woman or the dirt bike racer or both are now on this train in **IMAGE#2408549**, heading towards a new location for perhaps the next race or a well-deserved vacation post celebration. This train ride could represent their movement towards the next chapter in their journey, literally as well as metaphorically.

The passenger looking out the window could be contemplating the recent event of the dirt bike race victory and the warm celebration that followed. The train's vigorous motion and the reflected sunlight might be symbolising their sense of achievement and their eager anticipation for upcoming adventures. Thus, these images together weave a tale of triumph, celebration, and anticipation for what lies ahead.

Figure 16: Dialogue example from SparklesDialogueVG. Visual images were not provided to GPT-4 during data generation but will be incorporated during SparklesChat training.

```
7       {'role': 'assistant', 'content': 'In IMAGE#2408549, we witness the
            rush of a large, striking blue train speeding through a railroad
            crossing. A passenger, perhaps contemplating the journey,
            watches the world outside through a window. This train scene can
            signify a journey or transition.\\n\\nBringing these images
            together into a coherent story, it might be speculated that
            after the celebration in IMAGE#2331159 for the successful dirt
            bike race from IMAGE#2330601, the next chapter of their journey
            is starting. It could be speculated that the woman or the dirt
            bike racer or both are now on this train in IMAGE#2408549,
            heading towards a new location for perhaps the next race or a
            well-deserved vacation post celebration. This train ride could
            represent their movement towards the next chapter in their
            journey, literally as well as metaphorically.\\n\\nThe passenger
             looking out the window could be contemplating the recent event
            of the dirt bike race victory and the warm celebration that
            followed. The train's vigorous motion and the reflected sunlight
             might be symbolising their sense of achievement and their eager
```

```
                  anticipation for upcoming adventures. Thus, these images
                  together weave a tale of triumph, celebration, and anticipation
                  for what lies ahead.'}]
8    ]
9    ```
```

## J.2 MULTIPLE DIALOGUES GENERATION FOR SPARKLESDIALOGUECC

For SparklesDialogueCC, we prompt GPT-4 to generate three dialogues in a single response. These dialogues incorporate one, two, and three images in the first turn and a single image in the second. Each prompt includes three demonstration dialogues and nine candidate image descriptions to facilitate this. We curated 150 demonstration dialogues, evenly split with 50 dialogues for each type. The complete image-text dataset comprises about 3,500 pairs.

We first present our designed **prompt** for GPT-assisted Multiple Dialogues Generation to generate SparklesDialogueCC in Table 9. Then, we show a case of the `Dialogue Demonstrations` and `Candidate Image Descriptions` to construct the prompt. Finally, we show the corresponding **generated dialogues** using the example prompt.

**Example of dialogue demonstrations.** We visualize the images corresponding to image IDs in the dialogues in Figure 17 for reference, while these visual images were not sent to GPT-4 for data generation. Note that we abbreviate the message content of the assistant in the second turn as "..." to save space, considering that the previous message contents have provided enough demonstrations.

```
1    ```json
2    [
3    [
4        {'role': 'user', 'image_ids': ['3775'], 'content': 'What kind of ink
            is usually used for such tattoos that we see in IMAGE#3775, and
            how long can we expect it to last?'},
5        {'role': 'assistant', 'content': 'Tattoo artists typically use a
            specific type of ink known as tattoo ink for their designs, like
            the one shown in IMAGE#3775. Tattoo ink is usually made from
            metal salts, plastic-based pigments, or heavy metals for creating
            different colors, although black ink, like the one used in the "
            trust yourself" tattoo in the image, is generally carbon-based.\n
            \nHow long the tattoo lasts in its crisp form depends on a series
            of factors such as the depth of the ink placement, the type of
            ink used, how well it\'s cared for during the healing process,
            and how it\'s protected from the sun and aging. \n\nGenerally,
            tattoos are permanent because the human body\'s immune system isn
            \'t capable of dealing with the large molecules of tattoo ink.
            Over several years, a tattoo can fade slightly due to the natural
            regeneration cycle of skin cells, but it won\'t disappear
            completely. The good thing about black tattoos, as seen in IMAGE#
            3775, is that black pigment is absorbed well by all skin types
            and tends to last longer without fading. However, the specific
            longevity of a tattoo will differ from person to person. If one
            wants their tattoo\'s health to be maintained, they should keep
            it moisturized, use sunscreen when out in harsh sun, and consider
            touch-ups from a professional tattoo artist when necessary.'},
6        {'role': 'user', 'image_ids': ['4799'], 'content': 'Can the ink used
            in the tattoo from IMAGE#3775 potentially cause any skin
            irritations or allergies, similar to how some people have
            reactions to certain foods like in IMAGE#4799?'},
7        {'role': 'assistant', 'content': '...'}
8    ],
9    [
10       {'role': 'user', 'image_ids': ['301', '3122'], 'content': 'What kind
            of environment is depicted in IMAGE#301 and how does it contrast
            with the one in IMAGE#3122?'},
11       {'role': 'assistant', 'content': "IMAGE#301 depicts a tropical, beach
            environment. In the image, there are two kayaks resting on a
```

Table 9: Prompt for GPT-assisted multiple dialogues generation.

Users will interact with a conversational assistant that has advanced capabilities of understanding, analyzing, and reasoning about images. This includes discussing a variety of real-world concepts, objects, and entities, generating a range of text materials, seeking advice, guidance, or assistance, and much more.

Below are three illustrative dialogues presented in a JSON format. Each one represents a self-contained conversation between a "user" and the "assistant" regarding multiple images. Each "user" message contains an "image_ids" field recording the IDs of newly selected images. The images are referred to in the "content" field as IMAGE#image_id.

```json
{Dialogue Demonstrations}
```

Please note that the specific "image_ids" and "content" in the JSON above are for illustrative purposes only. The actual candidate images are shown below delimited by triple quotes, each accompanied by an image ID and a caption. Avoid using phrases similar to 'caption' and 'description' in your dialogue as if the user and the assistant have visual capabilities.

```json
{Candidate Image Descriptions}
```

Each dialogue consists of four messages:
1. A user examines all candidate images, selects highly relevant ones, and sends a reasonable and creative message to the assistant.
2. Once the images are provided, the assistant thoroughly perceives and comprehends them, responding with highly helpful and exceptionally detailed answers that provide comprehensive reasoning.
3. Considering the past dialogue, the user chooses another candidate image for further inquiry. The user should refer to both the newly selected image and those mentioned earlier in the same dialogue.
4. The assistant provides a highly helpful and exceptionally detailed answer providing comprehensive reasoning regarding the visual content of the images.

The following are three independent dialogues between the user and the assistant, adhering to the given JSON format. In this format, the first message in the three dialogues includes 1, 2, and 3 image IDs respectively.
Make sure to formulate accurate and diverse "content" that does not strictly follow the illustrative dialogues. And remember to develop the last "content" even though it is shown as "..." in the JSON format provided above.

---

Dialogue Demonstrations

```
[[{'role': 'user', 'image_ids': I^{1,1}, 'content': X_q^{1,1}},
  {'role': 'assistant', 'content': X_a^{1,1}},
  {'role': 'user', 'image_ids': I^{1,2}, 'content': X_q^{1,2}},
  {'role': 'assistant', 'content': '...'}],
 ......
 [{'role': 'user', 'image_ids': I^{3,1}, 'content': X_q^{3,1}},
  {'role': 'assistant', 'content': X_a^{3,1}},
  {'role': 'user', 'image_ids': I^{3,2}, 'content': X_q^{3,2}},
  {'role': 'assistant', 'content': '...'}]]
```

Candidate Image Descriptions

```
[{'image_id': J^1, 'caption': C^1},
 ......
 {'image_id': J^9, 'caption': C^9}]
```

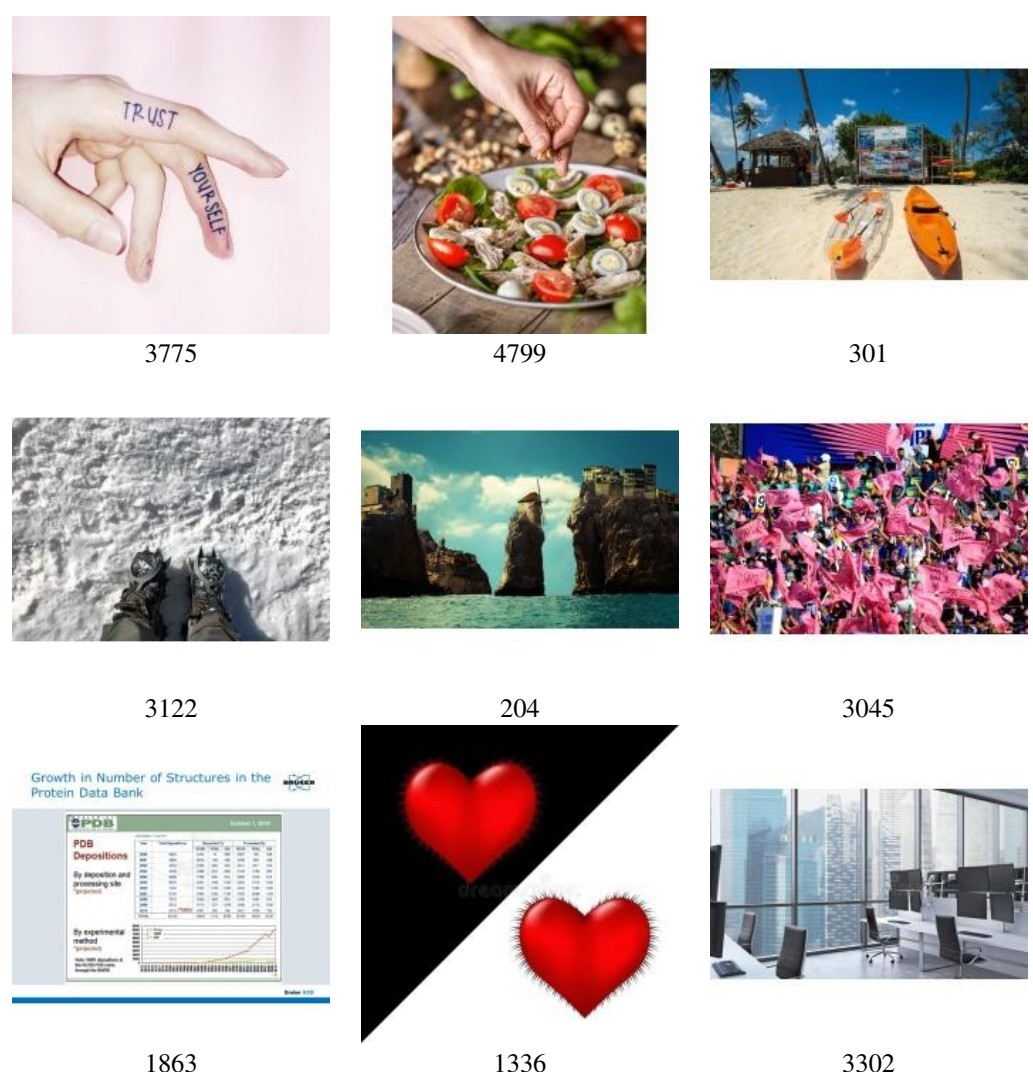

Figure 17: Reference images corresponding to the image IDs in the demonstration dialogues in section J.2. These images were not sent to GPT-4 for data generation.

```
sandy beach, and this detail hints at a location that's likely in
 a warmer climate, suitable for water sports and recreational
activities. The presence of palm trees further supports this, as
they are commonly found in regions that are hot and sunny most of
 the year. The blue sky and the small building that looks like a
beach hut or kiosk suggest a relaxed, leisure-oriented
environment.\n\nOn the contrary, IMAGE#3122 portrays a snowy,
mountainous environment. The presence of snow suggests a colder
climate, likely during winter. The pair of black rubber snowshoes
 with metal spikes on the bottom implies activities related to
snow, such as hiking or mountain climbing. The person wearing a
hooded jacket and grey pants with their hands in their pockets
further indicates the cold nature of this environment. The trees
and mountains visible through the snow complete the impression of
 a wilderness setting, possibly remote and less populated.\n\nIn
summary, the two images contrast significantly – one depicts a
warm, tropical beach setting that is likely oriented towards
leisure and relaxation, while the other depicts a colder,
mountainous setting associated with more strenuous outdoor
```

```
12      {'role': 'user', 'image_ids': ['204'], 'content': 'Does the scene in
            IMAGE#204 remind you of the environment depicted in IMAGE#301, or
            are they entirely different?'},
13      {'role': 'assistant', 'content': '...'}
14  ],
15  [
16      {'role': 'user', 'image_ids': ['3045', '1863', '1336'], 'content': '
            Generate a promotional text that could fit for a sports event
            based on IMAGE#3045. Also, please include elements from IMAGE#186
            3 and IMAGE#1336 to give it some unique angles.'},
17      {'role': 'assistant', 'content': "Join us for an electrifying display
             of sportsmanship like no other! Just as the fans in IMAGE#3045,
            cheer on your favorite team in an outdoor stadium filled with a
            roaring crowd, all decked out in their team's gear, waving pink
            flags in unity and excitement. Experience an atmosphere buzzing
            with energy beneath the open sky, surrounded by the vibrant
            greenery.\n\nDrawing inspiration from IMAGE#1863, watch as the
            team's performance graph soars high, much like the increasing
            deposits in a bank account. Witness the thrilling trajectory of
            action unfold, as each second, each decision, contributes to an
            upward trend in their game. Be a part of this journey, as they
            race not just to maintain the pace but to set a new norm, akin to
             the decreasing withdrawals mark, showcasing improvement and
            growth over time.\n\nIn the spirit of IMAGE#1336, let your hearts
             beat as one, with the passion of the red and the harmony of the
            black and white. Let the dual celebrations of vibrant love for
            the game and steadfast unity resound in the air. Let the duality
            of a heartbeat stir a symphony of balance - just like the yin and
             yang of competitiveness and sportsmanship.\n\nSo, let's delve
            into this exciting world, where not just the players, but even
            the spectators embark on a journey of passion, unity, growth, and
             harmony. Let the games begin!"},
18      {'role': 'user', 'image_ids': ['3302'], 'content': 'Could the
            atmosphere and energy captured in IMAGE#3045 somehow be invoked
            in the office space depicted in IMAGE#3302 to make it more lively
            ? Any suggestions?'},
19      {'role': 'assistant', 'content': '...'}
20  ]
21  ]
22  ```
```

**Example of candidate image descriptions.** An example of Candidate Image Descriptions is shown below, and their corresponding source images are shown in Figure 18 for reference (they are not sent to GPT-4).

```
1  ```json
2  [
3      {'image_id': '2439', 'caption': 'This image shows a kitchen with
            wooden cabinets, black countertops, and white appliances. The
            floor is made of tiles and the walls are painted white. There is
            a large window above the sink that lets in plenty of natural
            light. The room is spacious and well lit.'},
4      {'image_id': '3065', 'caption': "This is an image of an airplane
            flying in the sky at sunset. The plane is a large, commercial jet
             with a white body and red and blue stripes on the tail. It is
            flying low in the sky, with the sun setting behind it, casting a
            warm orange glow on the left side of the image and a blue glow on
             the right. The plane's engines are visible at the bottom of the
            image, with smoke coming from them. The sky is a deep blue, with
            clouds in the distance that are tinged with pink from the sunset.
            "},
```

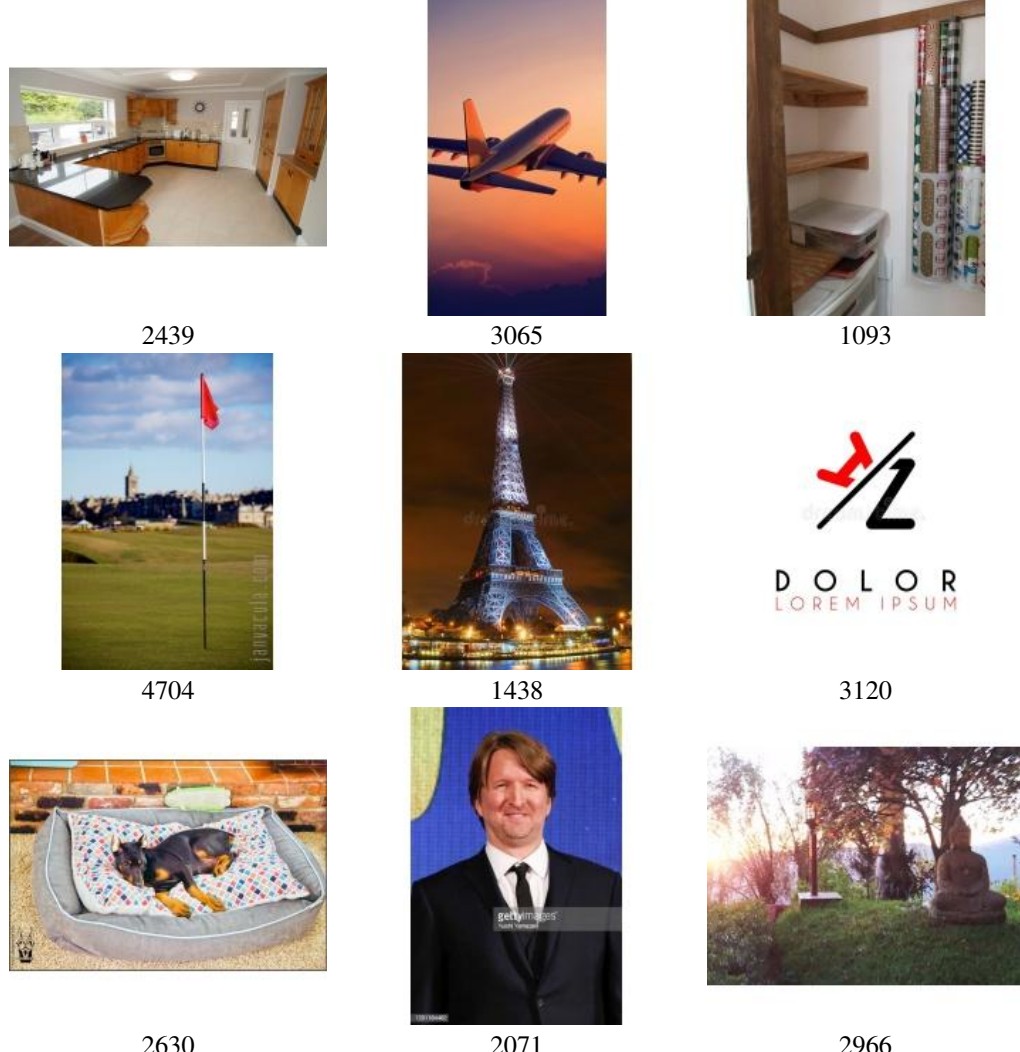

Figure 18: Candidate images corresponding to the image IDs in the dialogues generation process in section J.2. These images were not sent to GPT-4 for data generation.

```
5      {'image_id': '1093', 'caption': 'The image shows a small room with a
           wooden shelf on the wall, several rolls of wrapping paper stacked
            on it, a door on the right side, and a window on the left side.
           The walls are painted white and there is a wooden floor.'},
6      {'image_id': '4704', 'caption': 'The image shows a view of a golf
           course with a red flag on the green. In the background, there is
           a city skyline with buildings and a church steeple. The grass on
           the course is lush and green, and there are trees on either side
           of the fairway. The sky is clear and blue, and there are a few
           clouds in the distance. The flag on the green is a small, red
           flag with a white pole. It is standing upright in the middle of
           the green, and it looks like it is blowing in the wind. The city
           skyline in the background is quite impressive, with several tall
           buildings and a church steeple visible. The church steeple is
           quite tall and has a pointed top.'},
7      {'image_id': '1438', 'caption': 'The Eiffel Tower is a famous
           landmark in Paris, France. It is a wrought iron lattice tower
           that was built in 1889 to commemorate the centenary of the French
            Revolution. The tower stands 324 meters tall and is located on
```

```
           the Champ de Mars in the heart of Paris. It is one of the most
           visited tourist attractions in the world, with millions of people
            visiting it every year. The tower has become an iconic symbol of
            Paris and France. The tower is painted in blue, white, and red,
           the colors of the French flag. The lights on the tower create a
           beautiful and magical atmosphere, making it a popular spot for
           romantic walks and photos. The tower is surrounded by water, with
            a river running underneath it.'},
8       {'image_id': '3120', 'caption': 'This image is a logo for a company
           or brand. The logo consists of the letters "z" and "l" in a red
           and black color scheme. The letters are connected by a diagonal
           line through the center of the image, creating a simple and
           modern design suitable for various businesses and industries.'},
9       {'image_id': '2630', 'caption': "The image shows a black and tan dog
           lying on a large, gray dog bed. The bed has a geometric pattern
           on it and is placed in front of a brick fireplace. The dog's head
            is resting on its paws, and it appears to be sleeping. The image
            is well lit and the details of the dog and the bed are clear."},
10      {'image_id': '2071', 'caption': "The image shows a man in a black
           suit and tie standing in front of a blue and yellow background.
           The man is smiling and has his hands in his pockets.\n\nThe man
           is wearing a black suit and tie, with a white shirt underneath.
           He has short, dark hair and a well-groomed beard. He is standing
           in front of a large blue and yellow background that looks like a
           cartoon character's face. The character's eyes and mouth are wide
            open, and it looks like the man is standing in front of the
           character's face.\n\nThe background looks like a cartoon
           character's face, with blue and yellow colors. The character's
           eyes and mouth are wide open, and it looks like the man is
           standing in front of the character's face. The character's eyes
           and mouth are wide open, and it looks like the man is standing in
            front of the character's face."},
11      {'image_id': '2966', 'caption': 'The image shows a garden with a
           stone statue of a person sitting under a tree. The statue has a
           serene expression on its face, and the sun is setting behind the
           trees, casting a warm orange light on the scene. The foreground
           grass is green and well-maintained, and there are no other
           objects or people in the image.'}
12  ]
13  ```
```

**Dialogue examples from SparklesDialogueCC.** The generated dialogue is visualized in Figure 19. The raw text is shown as follows. The image IDs in the dialogues refer to the images in Figure 18.

```
1   ```json
2   [
3   [
4       {'role': 'user', 'image_ids': ['3120'], 'content': 'Could you tell
           me what the design of the logo in IMAGE#3120 might symbolize or
           represent?'},
5       {'role': 'assistant', 'content': 'The logo in IMAGE#3120, with the
           combined letters "z" and "l" connected by a diagonal line, seems
            to convey simplicity, modernity, and approachability. The use
           of "z" and "l" could represent the initials of a company or
           individual\'s name, while the diagonal line may symbolize a link
            or connection, emphasizing unity or collaboration. The choice
           of red and black colors could be conveying a sense of power and
           sleek sophistication, respectively. Black often symbolizes
           solidity and formality, while red can represent passion and
           energy. However, the specific meaning could vary as logo
           interpretations often depend on the organization\'s mission,
           brand values, and the industry it\'s part of.'},
6       {'role': 'user', 'image_ids': ['1438'], 'content': 'Considering the
           design and message one might infer from the logo in IMAGE#3120,
```

Generate a promotional text that could fit for a sports event based on IMAGE#3045

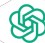 . Also, please include elements from IMAGE#1863 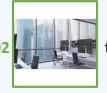 and IMAGE#1336

to give it some unique angles.

Join us for an electrifying display of sportsmanship like no other! Just as the fans in IMAGE#3045, cheer on your favorite team in an outdoor stadium filled with a roaring crowd, all decked out in their team's gear, waving pink flags in unity and excitement. Experience an atmosphere buzzing with energy beneath the open sky, surrounded by the vibrant greenery.

Drawing inspiration from IMAGE#1863, watch as the team's performance graph soars high, much like the increasing deposits in a bank account. Witness the thrilling trajectory of action unfold, as each second, each decision, contributes to an upward trend in their game. Be a part of this journey, as they race not just to maintain the pace but to set a new norm, akin to the decreasing withdrawals mark, showcasing improvement and growth over time.\n\nIn the spirit of IMAGE#1336 let your hearts beat as one, with the passion of the red and the harmony of the black and white. Let the dual celebrations of vibrant love for the game and steadfast unity resound in the air. Let the duality of a heartbeat stir a symphony of balance - just like the yin and yang of competitiveness and sportsmanship.

So, let's delve into this exciting world, where not just the players, but even the spectators embark on a journey of passion, unity, growth, and harmony. Let the games begin!

Could the atmosphere and energy captured in IMAGE#3045 somehow be invoked in the

office space depicted in IMAGE#3302 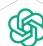 to make it more lively? Any suggestions?

Absolutely! The outdoor stadium setting of IMAGE#3045 exudes an energy and unity that can certainly be incorporated into IMAGE#3302, not in a literal sense of course, but more so in creating an ambiance and a sense of togetherness and excitement in the workspace.

For starters, we can draw on the crowd energy and unity depicted in IMAGE#3045. Adding some group oriented spaces in the office, like a collaboration zone with comfortable seating, can be a great way of fostering teamwork and unity - much like the fans watching a game together. A big screen showing motivational or industry related dynamic content like live news, updates, or even sports in this area could spark excitement and conversations, stimulating a dynamic work environment.

The greenery in the stadium can be mirrored in the office by including indoor plants. This can bring a touch of nature inside, invoking a sense of calm yet vibrant energy, making the space inviting and friendly.

IMAGE#3045 has fans displaying their team's colors in form of flags and jerseys. Similarly, using brand colors in the office space can evoke a sense of pride and connection to the company's mission. This can even be extended by having artwork or banners that tell the company story or milestones, resembling the signs and banners held up by fans in the image.

From IMAGE#1863, the upward trend of the graph suggests progress and growth. This theme can be maintained in the office by showcasing company progress visuals, awards, or integrated performance dashboards to highlight team victories and achievements, creating a sense of accomplishment among employees.

Taking cues from IMAGE#1336, the office interior can imply balance and harmony. The red heart corresponds to spaces or elements designed for enthusiastic brainstorming and vibrant energy, whereas areas symbolizing the black and white heart can be designed for more focused, calm, and harmonious work.

By blending elements from all three images, the office in IMAGE#3302 can indeed be transformed into a place that is energetic, lively, motivating, while also comforting and balanced - a true field of productivity.

Figure 19: Dialogue example from SparklesDialogueCC. Visual images were not provided to GPT-4 during data generation but will be incorporated during SparklesChat training.

```
                 would it be a good fit for the place depicted in IMAGE#1438? How
                    would they complement each other?'},
7        {'role': 'assistant', 'content': 'The logo from IMAGE#3120 can bring
                    a modern, energetic, and connected vibe to the traditional and
                    iconic setting of the Eiffel Tower in IMAGE#1438. This can
                    create an interesting juxtaposition between historic and
                    contemporary themes.\n\nThe simplicity and modernity of the logo
                    design would sharply contrast yet harmonize with the complexity
```

```
                and intricacy of the Eiffel Tower. This could appeal to
                audiences who appreciate both the historical and futuristic
                aspects of a brand. Similarly, the vibrancy and energy indicated
                 by the red color in the logo may align well with the romantic
                and lively atmosphere associated with Paris, the city where
                Eiffel Tower is situated.\n\nIf this was for a brand that
                intended to evoke a sense of modern connection in a place
                steeped with history, or perhaps a company that combines
                technology with tourism in Paris, this logo could indeed
                complement the place effectively.'}],
 8  [
 9        {'role': 'user', 'image_ids': ['2630', '3065'], 'content': 'Can you
                imagine a story that might link the peaceful dog lounging in
                IMAGE#2630 and the airplane soaring in the sunset sky of IMAGE#3
                065?'},
10        {'role': 'assistant', 'content': 'Certainly! Here\'s a story
                possibility: The dog in IMAGE#2630, named Boomer, belongs to a
                dedicated pilot who often takes on long haul flights, like the
                airplane depicted in IMAGE#3065. Whenever the pilot sets off for
                 a lengthy trip, Boomer retreats to his comfortable dog bed in
                front of the cozy fireplace, waiting for his owner\'s return.\n\
                nThe pilot, while often away, never fails to think about Boomer.
                 From high above the earth, amidst beautiful sunsets, the canine
                \'s owner counts the hours until they can be reunited. Just the
                thought of Boomer's serene expression on the tastefully
                patterned dog bed provides a sense of peace and comfort even
                thousands of miles away.\n\nSo despite their geographical
                separation depicted in these images, there's an underlying
                connection of loyalty and companionship between Boomer and his
                pilot owner, which is a heartwarming staple in their world
                filled with frequent departures and heartfelt reunions.'},
11        {'role': 'user', 'image_ids': ['4704'], 'content': 'How would the
                atmosphere in IMAGE#4704 contribute to the story of our pilot-
                owner and Boomer from IMAGE#2630 and IMAGE#3065?'},
12        {'role': 'assistant', 'content': 'The setting in IMAGE#4704 could
                introduce another layer to our narrative. Imagine this: When off
                -duty, our pilot-owner is also an avid golfer who cherishes the
                peacefulness and serenity found on the golf course, like the one
                 seen in IMAGE#4704. After returning from a long trip, the pilot
                 usually heads to the fairway, and Boomer eagerly accompanies
                him.\n\nThe manicured greens serve as a perfect place for Boomer
                 to roam freely, chase golf balls, and feel the wind in his fur.
                 The distant city skyline seen from the golf course acts as a
                reminder of the urban life our pilot leaves behind when they
                find solace in the peaceful embrace of the game and their
                cherished canine companion.\n\nSubsequently, the lush green golf
                 course fills in a missing piece of the puzzle, acting as a
                mediating space that connects the frequent travels of the pilot
                and Boomer\'s waiting. It becomes a place of joy, reconnection,
                and shared experiences between the duo, marking the happy ending
                 to their waiting and the start of their time together until the
                 next flight.'}],
13  [
14        {'role': 'user', 'image_ids': ['2439', '1093', '2071'], 'content': '
                Could you help me design a scenario the place in IMAGE#1093 is
                being used as a temporary office for the man in IMAGE#2071, who
                usually works in the kitchen in IMAGE#2439?'},
15        {'role': 'assistant', 'content': 'Sure, here\'s a possible scenario:
                In IMAGE#2071, we see a man who typically works from his spacious
                 kitchen shown in IMAGE#2439. The kitchen, with its wooden
                cabinets and black countertops, doubles as his office. Its large
                windows provide ample natural light, creating a conducive and
                productive work environment. It is here that he plans his
                strategies, makes important business decisions, and leads virtual
                 meetings.\n\nHowever, for a change of scenario - perhaps due to
```

```
         a home renovation or family visiting – the man decides to shift
         his workspace to the small room shown in IMAGE#1093 for a while.
         The room is likely a storage space or craft room with wrapping
         paper stacked on a wooden shelf, indicating creativity and
         resourcefulness.\n\nHe sets up his temporary office here,
         positioning his desk near the window to replicate the natural
         light he enjoys in his kitchen-office. The surrounding array of
         vibrant wrapping paper inspires creativity and brings a touch of
         liveliness to his work routine. Also, the closed door ensures
         that he has quiet and privacy for his business calls.\n\nSo in
         this unusual but charming workspace, our businessman continues
         his work, exhibiting that all you need to be productive is a bit
         of adaptability, irrespective of where you are.'},
16       {'role': 'user', 'image_ids': ['2966'], 'content': 'Taking into
         account all the previous images, how could we incorporate the
         scene from IMAGE#2966 into the story of the man changing his
         workspace?'},
17       {'role': 'assistant', 'content': 'The serene garden scene in IMAGE#29
         66 could be an excellent addition to the narrative. Let\'s say,
         towards the back of the house where both the kitchen-office and
         the temporary workspace reside, there's this quiet garden
         depicted in the image, featuring a stone statue sitting under a
         tree.\n\nAt the end of each workday, the man enjoys taking a
         stroll in the garden to unwind. The setting sun casts a warm
         orange light on the scene creating a peaceful atmosphere, and he
         often finds himself sitting next to the stone statue, reflecting
         on his day. The tranquility of the garden effectively helps him
         transition from work mode, and it provides him with a sense of
         balance that complements his adaptable work life.\n\nTherefore,
         despite the change in his workspace, the man maintains his
         evening ritual in the garden, capturing the idea that while our
         surroundings may change, there's comfort in keeping certain
         routines and finding peace in nature.'}
18   ]
19   ]
20   ```
```

## K   DIALOGUE DEMONSTRATIONS' CONTRIBUTION TO DATA QUALITY AND DIVERSITY

`Dialogue Demonstrations` act as contextual learning examples, guiding GPT-4 to produce responses that are both well-formatted and diverse. To demonstrate this effect, we modified the original **prompt** used for GPT-assisted Multiple Dialogues Generation, as detailed in Table 9, by removing content relating to demonstration dialogues. We then employed the same `Candidate Image Descriptions` as in Figure 15 to create a new prompt and generate a response. The resulting response was inferior in quality, failing to meet the desired formatting criteria, such as assigning image IDs, specifying the number of images per dialogue turn, and incorporating new images in subsequent turns. Furthermore, the response lacked the diversity that its dialogues typically ask for more detailed descriptions of images but not specifying particular aspects. In conclusion, dialogue demonstrations are crucial not only for enhancing data quality by providing formatting guidelines but also for increasing diversity by conditioning different demonstrations.

**Dialogue 1:**

1. **User:** I'm working on a home renovation project and need some kitchen inspiration. The image with ID 2439 caught my eye; could you give a more detailed account of it?

2. **Assistant:** Absolutely, I'd be happy to! The kitchen space shown in image 2439 exhibits a clean and bright style, with an airy feel. The cabinets, constructed of wood, have an earthy

and natural charm. They complement the black countertops well, adding a layer of contrast. The appliances are white, offering a sense of cleanliness and simplicity. The floor is tiled, a practical choice for a room that often grapples with spills and frequent foot traffic. A substantial window draws in an ample amount of natural light, brightening the room and making it appear spacious. The white paint on the walls enhances this light, reflecting it to confer an even more airy ambience. This type of kitchen design is both efficient and welcoming, making it an ideal space for cooking and entertaining.

3. **User:** Great, that was very helpful! One more thing - is there anything in this kitchen that could be improved or updated?

4. **Assistant:** Well, any suggestions for improvement would largely depend on your personal style and functionality needs. However, here are a couple of general ideas: Whilst the window lets in a large amount of natural light, at night the room might need more illumination. Consider upgrading the lighting fixtures for a brighter and more evenly lit kitchen after sunset. Depending on the design scheme, you could select modern pendant lights, under-cabinet lights, or recessed lights. Additionally, the countertops appear to be clear of clutter, which is great, but there doesn't seem to be much in the way of storage solutions on them. Maybe you could use some stylish canisters for everyday utensils or ingredients, providing storage while keeping items within easy reach. Remember, home renovation is personal and subjective, so it's always a good idea to take inspiration from different sources, but ultimately stick to what will make you and the other users of the kitchen most content and comfortable.

---

**Dialogue 2:**

1. **User:** I'm curious about the photos with IDs 3065 and 2630. Could you explain these in more detail?

2. **Assistant:** Sure thing! In image 3065, we see an impressive dramatization of human achievement and the beauty of nature. It captures a large commercial airplane in the midst of flight under a sunset. The aircraft, distinguished by a white body and red and blue stripes on the tail, is low flying in the sky. Rumbling engines can be seen at the bottom, belching smoke. The setting sun irradiates the left side of the image with a warm, orange glow, while the right side basks in a cooler hue of blue. The sky's depth is accentuated by its rich blue tones, and the distantly scattered pink-tinged clouds add a touch of romanticism to the scene. Image 2630 is an adorable and intimate snapshot of a companion animal, a black and tan dog lying on a sizable gray dog bed. The bed's geometric pattern and substantial size suggest its comfort. Located before a brick fireplace, the setting seems cozy and warm. It appears as if the dog is sleeping, its snout resting atop its folded front legs. The clarity and lighting in the image allow us to appreciate the serene and comfortable atmosphere it represents.

3. **User:** I find those details on image 3065 very helpful. I'm working on an art project that involves illustrating a story about a family dog. How can I incorporate elements from image 2630 into a scene involving a plane flying at sunset as in image 3065?

4. **Assistant:** That sounds like a captivating art project! If you're trying to harmonize both images into one scene, here are a few suggestions: You could keep the breathtaking sunset sky with the airplane from image 3065 as the background. It brings a sense of vastness and adventure. However, instead of showing the airplane up in the sky, you could bring it closer to the ground, perhaps just made it after a landing or just before taking off on a runway near the home with the dog. In the foreground, you could portray a homely scene inspired by image 2630. The dog could be lying on its comfy bed, looking up at the airplane in the sky with curiosity, or perhaps excitement. To create a strong emotional connection, the placement of the dog and its bed could be strategic, maybe by a window that shows the plane in the sky, demonstrating that the two scenes are connected. Finally, you might want to add elements that link the dog to the plane. Perhaps a family member waving goodbye in the distance near the plane, or a toy plane near the dog to show the connection between the domestic scene and the broader story. This could convey notions of longing, anticipation, or excitement.

**Dialogue 3:**

1. **User:** I've been assessing images 1438, 2630, and 3120 and would like some further clarification on them.

2. **Assistant:** Of course, I'd be happy to help. In image 1438, we see the Eiffel Tower, an international symbol of Paris and France, standing tall. It's painted in blue, white, and beneath it runs a river. The tower's lights create a magical ambience that has made it a popular site for strolls and photos. Image 2630 depicts a peaceful scene of a black and tan dog sleeping on a large, gray dog bed. The bed, sporting a geometric pattern, is placed in front of a brick fireplace, amplifying the homely and comfort vibes the picture transmits. Image 3120 shows a distinctive logo for a company or brand. The logo comprises a 'z' and an 'l' in a red and black color scheme. The letters are uniquely connected by a diagonal line running through the image's center, rendering a modern and minimal design suitable for various businesses.

3. **User:** I'm starting a pet-friendly accommodation hosting service in Paris and trying to come up with a logo. Could I blend elements from the images of the Eiffel Tower and the dog into the logo design like in image 3120?

4. **Assistant:** Combining those elements sounds like a great idea! Here's a suggestion: Use the 'z' and 'l' shape from the logo in image 3120, but perhaps integrate elements that signify your business's unique selling point, being pet-friendly and located in Paris. You could morph the 'z' into a simplified, abstract version of the Eiffel Tower, keeping the minimal and sharp edges. On the other hand, the 'l' could be transformed into an abstract, stylized rendition of a dog, keeping the balance between simplicity and recognition. To make it clear that the accommodation is pet-friendly, consider incorporating paw prints as embellishments or as a background pattern. The color scheme of red, white, and blue could be another subtle nod to your location in France, inheriting the theme from the Eiffel Tower's image. Ultimately, the design should resonate with your target audience, so using familiar symbols (like the Eiffel Tower and a dog) in a clean, modern design could effectively communicate your brand's purpose and values.

