# OpenReview forum: "Sparkles: Unlocking Chats Across Multiple Images for Multimodal Instruction-Following Models"
_ICLR.cc/2024/Conference — Submitted to ICLR 2024_

### Official Review · Reviewer_qmnH · 2023-10-28

**Soundness:** 3 good
**Presentation:** 3 good
**Contribution:** 3 good
**Rating:** 6
**Confidence:** 4

**Summary:**

This work studies the open multimodal dialogue following user instruction in the conversations of multiple turns with multiple images. This work achieves this from three directions by proposing a model (SparklesChat), a dataset (SparklesDialogue), and a benchmark (SparklesEval). It also performs experiments on SparklesEval, the BISON binary image selection, and the NLVR2 visual reasoning task, on which SparklesChat outperforms MiniGPT-4 significantly.

**Strengths:**

1. This is one of the first works that studies multiple turns with multiple images for the open multimodal dialogue. Thus, it can additionally evaluate cross-image and cross-turn coherence and completeness of responses.

2. It contributes a novel dataset named SparklesDialogue leveraging GPT-4.

3. This work also proposes GPT-assisted evaluation named SparklesEval that can automate quantitative evaluation of a model’s conversation across multiple images and dialogue turns.

4. The Appendix and the supplementary material is helpful and very thorough.

**Weaknesses:**

1. It only consider two images per context, which could be too structure with little diversity.

2. The SparklesChat model is not novel in that it is just an instruction tuned miniGPT-4. It could be removed from contributions.

3. As described in Table 1, SparklesDialogue is not large-scale.

4. Each conversation seems to have a very typical pattern with two images as described in section 2.
“In the first turn, the user initiates a reasonable and creative message regarding some images. In response, the assistant generates detailed answers that include comprehensive reasoning regarding the visual content. In the second turn, the user introduces a new image for further discussion, referencing both the new and previous images.”

5. Only the miniGPT-4 is compared as a baseline.

**Questions:**

1. Why only the two datasets - BISON and NLVR2 are chosen? Is there any other dataset to use?

---

> ### Author Response · Authors · 2023-11-23
> **Response to Reviewer qmnH**
>
> We deeply appreciate your comprehensive review and are grateful for your positive remarks about our novel dataset, benchmark, and the thoroughness of our supplementary materials. We acknowledge your concerns and would like to address them and your question in the following response.
>
> ---
>
> **1: Regarding Image Numbers and Dataset Diversity**
>
> As shown in Table 1, our dataset spans one to four images across one or two dialogue turns and is not limited to two images. Additionally, the model can generalize to scenarios with more than four images or two turns. Additionally, **we include examples with four or five images in Appendix F of the revised manuscript to demonstrate this expanded capability.** We acknowledge your point on diversifying conversation patterns and plan to introduce additional templates and user-initiated conversation structures to expand the range of dialogue scenarios.
>
> **2: Clarifying Model Contribution**
>
> Our contribution to the model is the open-source SparklesChat to support chats across multiple images and dialogue turns. This capability has been a frequent request in issue trackers that remain unresolved since the release of popular multimodal instruction-following models such as MiniGPT-4 and LLaVA (see [issues 1](https://github.com/haotian-liu/LLaVA/issues/197), [2](https://github.com/Vision-CAIR/MiniGPT-4/issues/180), and [3](https://github.com/Vision-CAIR/MiniGPT-4/pull/232). We believe SparkelsChat is contributing to the community by pushing the boundaries of existing architectures to address novel challenges.
>
> **3: Scale of SparklesDialogue Dataset**
>
> The instructional capability of SparklesChat primarily stems from an instruction-tuned LLM (Vicuna). SparklesDialogue is designed to provide dialogue data to learn alignments involving multiple images and turns. With a total of 12K dialogue turn samples (4.5K x 2 + 2K x 2), our dataset is more than triple the size of MiniGPT-4's fine-tuning set (3.5K image-description pairs). We are committed to further expanding the dataset and enhancing its diversity to serve as a training and evaluation resource for multimodal dialogue models.
>
> **4: Broadening Baseline Models**:
>
> SparklesChat is built upon the MiniGPT-4 and adapted to accommodate multiple images due to its relevance and prominence in the field. We **extended our approach to more advanced models such as LLaVA-v1.5**to offer a broader view of our approach's effectiveness. LLaVA-v1.5 has improved its image input resolution from 224 pixels to 336 pixels. It is trained on a diverse dataset comprising about 665K instruction-following data. Additionally, LLaVA-v1.5* is fine-tuned on SparklesDialogue (about 6.5K) using the low-resource technique LoRA to conserve memory and time. The results are shown below and have **updated in the revised manuscript in Section 5.1 and Table 2.**
>
> | Model        | Instruction Data| BISON | NLVR2 | SparklesEval |
> |--------------|-----------------|-------|-------|-------|
> | GPT-4        | -               | -     | -     | 9.26  |
> | MiniGPT-4    | description     | 46.0% | 51.3% | 3.91  |
> | MiniGPT-4*   | description     | 51.3% | 46.7% | 3.50  |
> | LLaVA-v1.5   | Mixture (665K ) | 52.7% | 53.3% | 2.75  |
> | LLaVA-v1.5*  | **+SparklesDialogue**| 65.3% | 56.7% | 7.93  |
> | SparklesChat | description     | 52.0% | 48.0% | 3.06  |
> | SparklesChat | reasoning       | 52.7% | 54.0% | 6.71  |
> | SparklesChat | SparklesDialogue| 56.7% | 58.0% | 8.56  |
>
> According to the results, despite LLaVA-v1.5's advantages of higher image resolution (336 vs. 224 pixels) and a larger training set (665K vs. 6.5K), SparklesChat significantly outperforms LLaVA-v1.5 in three evaluation sets involving multiple images. While LLaVA-v1.5 outperforms MiniGPT-4 on BISON and NLVR2, it shows weaker results on SparklesEval. This may be due to LLaVA-v1.5's training data primarily focusing on closed-set multimodal tasks such as VQA, TextCaps, and RefCOCO, while lacking in open-ended dialogue training. After fine-tuning with our SparklesDialogue, LLaVA-v1.5* not only significantly improved in open-ended dialogue tasks but enhanced traditional multimodal tasks. These results validate the adaptability of our method in unlocking chats across multiple images for multimodal instruction-following models with minimal additional training cost.
>
>
> **5: Choice of Evaluation Dataset such as BISON and NLVR2**
>
> NLVR and BISON were selected for their use of natural images and the feasibility of automatic evaluation with unambiguous answers (e.g., TRUE/FALSE, single-choice problems). Based on your feedback, we aim to include broader evaluation benchmarks, incorporating images from diverse domains (e.g., documents, fashion, cartoons) and a wider range of evaluation metrics beyond accuracy measures.

---

### Official Review · Reviewer_eU8x · 2023-10-30

**Soundness:** 2 fair
**Presentation:** 3 good
**Contribution:** 1 poor
**Rating:** 5
**Confidence:** 4

**Summary:**

The paper introduces SparklesChat, a multimodal instruction following model for open-ended dialogues across multiple images. This is MiniGPT4 fine-tuned on the machine-generated dialogue dataset released in the paper called SparklesDialogue. This contains word-level interleaved multi-image and text interactions with up to 3 images during the first turn and 1 image during the second turn. SparklesDialogue consists of two subsets: 1) SparklesDialogueCC which contain images from CC3M and captions generated by MiniGPT4 2) SparklesDialogueVG which contain images from Visual Genome and descriptions from GPT-4, based on human-annotated captions, objects, and regions. SparklesEval is a new GPT-assisted benchmark with 150 dialogs, introduced to assess conversational competence across multiple images and dialogue turns, through criteria such as Image Understanding & Reasoning, Cross-Image & Cross-Turn Coherence, and Relevance & Completeness of Responses. SparklesChat outperforms MiniGPT-4 and gets marginally close GPT-4 on binary image selection task and the NLVR2 visual reasoning task. The paper contains ablation study on the effect of dialog turns and SparklesDialogue subsets during training.

====

Updated final rating form 3 to 5 due to demonstration of improvement on LLaVA as well. The experiments section and total contributions are still weak.

**Strengths:**

1.	New dataset SparklesDialogue for word-level interleaved multi-image and text interactions
2.	New benchmark SparklesEval for word-level interleaved multi-image and text interactions
3.	Demonstration of improved performance over MiniGPT4

**Weaknesses:**

1.	SparklesDialogue contains subset SparklesDialogueVG, which was generated using GPT-4. The paper compares with performance of GPT-4 (method used to create the data set is also being evaluated on), while still performing worse although SparklesChat uses much richer image embedding.
2.	No contribution in terms of novelty architecture. Main contribution is in the data set.
3.	Only two turns per sample in the dataset. Longer sessions are probably more practical than more images per turn and limiting to just 2 turns. Dataset (that too, machine-generated) being the highlight of this paper, would have expected more.
4.     Not clear how this extends to other approaches such as LLaVA. Results are shown only for Min-GPT4 extension.

**Questions:**

Q1) Section 5.2 mentions, SparklesDialogueVG and SparklesEval use the same sources of images and captions. This is suspected to be one of the reasons why model trained on SparklesDialogueVG performs better than model trained on SparklesDialogueCC. Isnlt this a serious issue, especially since SparklesDialogueVG is claimed to be the high quality subset?

Minor typo
1.	Table 2: Column title should be A2 under “Turn two”

**Details Of Ethics Concerns:**

The paper used GPT4 and MiniGPT-4 to create the datasets without any human review. It is unclear how safe the dataset is. Also, not clear about RAI.

---

> ### Author Response · Authors · 2023-11-23
> **Response to Reviewer eU8x**
>
> Thank you for your thorough review and the valuable insights! We appreciate your recognition of our new dataset and benchmark. In response to your concerns, we would like to provide the following clarifications and address the questions raised.
>
> ---
>
> **1. Comparison with GPT-4**
>
> SparklesChat underperforms compared to GPT-4, despite using richer image representations, because GPT-4 operates based on detailed image annotations and the evaluation set was generated by GPT-4. This gives GPT-4 an inherent advantage in image understanding tasks, as it does not face the same challenges in interpreting images.
>
> **2. Model Contribution**
>
> Thank you for pointing out that the dataset is our main contribution. Our contribution to the model is the open-source SparklesChat to support chats across multiple images and dialogue turns. This capability has been a frequent request in issue trackers that remain unresolved since the release of popular multimodal instruction-following models such as MiniGPT-4 and LLaVA (see [issues 1](https://github.com/haotian-liu/LLaVA/issues/197), [2](https://github.com/Vision-CAIR/MiniGPT-4/issues/180), and [3](https://github.com/Vision-CAIR/MiniGPT-4/pull/232)). We believe SparkelsChat is contributing to the community by pushing the boundaries of existing architectures to address novel challenges.
>
> **3. Adapt to More than Two-Turn Dialogue**
>
> We initially created two-turn dialogues to validate our ideas in unlocking chats across multiple images and turns, which is adaptable to more turns, as shown in Figure 8. Additionally, we acknowledge the practicality of longer sessions and plan to expand the dataset accordingly. This can be achieved by incorporating more turns of conversation based on the current two-turn dialogues or exploring new scenarios for images and text from diverse domains (e.g., documents, fashion, cartoons).
>
> **4. Extension to Other Approaches such as LLaVA**
>
> In light of your feedback, we **extended our approach to more advanced models such as LLaVA-v1.5**to offer a broader view of our approach's effectiveness. LLaVA-v1.5 has improved its image input resolution from 224 pixels to 336 pixels. It is trained on a diverse dataset comprising about 665K instruction-following data. Additionally, LLaVA-v1.5* is fine-tuned on SparklesDialogue (about 6.5K) using the low-resource technique LoRA to conserve memory and time. The results are shown below and have **updated in the revised manuscript in Section 5.1 and Table 2.**
>
> | Model        | Instruction Data| BISON | NLVR2 | SparklesEval |
> |--------------|-----------------|-------|-------|-------|
> | GPT-4        | -               | -     | -     | 9.26  |
> | MiniGPT-4    | description     | 46.0% | 51.3% | 3.91  |
> | MiniGPT-4*   | description     | 51.3% | 46.7% | 3.50  |
> | LLaVA-v1.5   | Mixture (665K ) | 52.7% | 53.3% | 2.75  |
> | LLaVA-v1.5*  | **+SparklesDialogue**| 65.3% | 56.7% | 7.93  |
> | SparklesChat | description     | 52.0% | 48.0% | 3.06  |
> | SparklesChat | reasoning       | 52.7% | 54.0% | 6.71  |
> | SparklesChat | SparklesDialogue| 56.7% | 58.0% | 8.56  |
>
> According to the results, despite LLaVA-v1.5's advantages of higher image resolution (336 vs. 224 pixels) and a larger training set (665K vs. 6.5K), SparklesChat significantly outperforms LLaVA-v1.5 in three evaluation sets involving multiple images. While LLaVA-v1.5 outperforms MiniGPT-4 on BISON and NLVR2, it shows weaker results on SparklesEval. This may be due to LLaVA-v1.5's training data primarily focusing on closed-set multimodal tasks such as VQA, TextCaps, and RefCOCO, while lacking in open-ended dialogue training. After fine-tuning with our SparklesDialogue, LLaVA-v1.5* not only significantly improved in open-ended dialogue tasks but enhanced traditional multimodal tasks. These results validate the adaptability of our method in unlocking chats across multiple images for multimodal instruction-following models with minimal additional training cost.
>
> **5. Similarity of Sources in SparklesDialogueVG and SparklesEval**
>
> Although there are similarities in image sources between SparklesDialogueVG and SparklesEval, the test set features unique combinations (e.g., 2-2 image pairings not present in training). The test set is carefully curated to ensure diversity, allowing for a comprehensive evaluation of multi-image dialogue capabilities. We acknowledge the similarity in image sources as a concern and plan to introduce additional data sources to provide a more balanced assessment.
>
> **6. Ethics Concerns**
>
> Our dataset and evaluation process has involved initial human review, including manual initiation and selection of dialogue demonstrations and rigorous quality checks, as detailed in Section 4.1 of our paper. We acknowledge the importance of thorough human review to ensure the safety and ethical soundness of the data.
>
> **7. Correction Acknowledged**
>
> Thank you for pointing out the typo. We have corrected it.

---

### Official Review · Reviewer_viMW · 2023-10-30

**Soundness:** 3 good
**Presentation:** 3 good
**Contribution:** 3 good
**Rating:** 6
**Confidence:** 3

**Summary:**

This paper presents SparklesChat, a multimodal instruction-following model for open-ended dialogues across multiple images. It introduces SparklesDialogue, a specialized machine-generated dialogue dataset, and achieves superior performance compared to MiniGPT-4 on vision-language benchmarks. SparklesChat's effectiveness is further demonstrated by its high score on SparklesEval, a benchmark for assessing conversational competence across multiple images and dialogue turns.

**Strengths:**

1. The paper addresses a key limitation in the field by introducing SparklesChat, a multimodal instruction-following model that integrates multiple images at the word level. This fine-grained integration of images and text is a novel approach that mimics natural human communication more closely.

2. The paper presents SparklesDialogue, the first machine-generated dialogue dataset designed for word-level interleaved multi-image and text interactions. The dataset is constructed from different image and description sources, ensuring greater robustness and diversity. Additionally, the paper introduces SparklesEval, a comprehensive scoring system that quantitatively evaluates the model's conversational competence in multimodal, open-ended dialogues.

3. The SparklesEval benchmark shows that SparklesChat's conversational competence significantly surpasses MiniGPT-4 and approaches the performance of GPT-4. These results highlight the potential of SparklesChat in real-world scenarios.

**Weaknesses:**

Considering the current status of single-image comprehension, which still requires further advancements, it appears that addressing scenarios involving multiple images may not be an immediate priority. Additionally, when considering the data construction approach described in the paper, it becomes evident that the model's capabilities are still constrained by the limitations of single-image understanding.

In my personal opinion, focusing on improving single-image comprehension would be more beneficial at this stage. Once single-image understanding is well-established, the demonstrated ability to handle multiple images, as showcased in the paper, should not pose significant challenges. It is crucial to ensure a solid foundation in single-image comprehension before delving into more complex scenarios involving multiple images.

**Questions:**

1. How do the Dialogue Demonstrations contribute to the data quality and diversity?
2. Considering the impressive performance of GPT-4 with ground truth (gt) annotation, could the authors provide a baseline using a strong caption model with an instruction-tuned Language Model to address the challenges raised in the paper?
3. Does the model in the paper have the capability to handle scenarios with more than two images, considering that the paper only showcases examples with two images?

---

> ### Author Response · Authors · 2023-11-23
> **Response to Reviewer viMW**
>
> Thank you for your insightful and constructive review of our work. We are encouraged by your recognition of the novelty and potential of our approach. We would like to address your concerns regarding the prioritization of single-image versus multi-image comprehension and other aspects related to our dataset and model capabilities.
>
> ---
>
> 1. **Prioritization of Single vs. Multi-Image Comprehension**
>     - We acknowledge the importance of single-image understanding in visual comprehension. However, our focus on multi-image dialogues stems from real-world complexity, where individuals often encounter and discuss multiple images simultaneously.
>     - Since the release of existing multimodal models such as MiniGPT-4 and LLaVA, there have been requests for multi-image capabilities in issue trackers that remain unresolved (see [issues 1](https://github.com/haotian-liu/LLaVA/issues/197), [2](https://github.com/Vision-CAIR/MiniGPT-4/issues/180), and [3](https://github.com/Vision-CAIR/MiniGPT-4/pull/232)). Our work aims to address these demands with our dataset and methodology.
>     - What's more, advancing multi-image dialogue capabilities complements single-image comprehension. Tackling complex scenarios with multiple images helps uncover limitations in single-image understanding and drives the field forward.
>
> 2. **Dialogue Demonstrations' Contribution to Data Quality and Diversity**:
>     - Thank you for your question! We **add examples and analysis in Appendix K**, to show that Dialogue Demonstrations act as contextual learning examples, guiding GPT-4 to produce well-formatted and diverse responses.
>     - To demonstrate this effect, we modified the original prompt used for GPT-assisted Multiple Dialogues Generation, as detailed in Table 9, by removing content relating to demonstration dialogues. We then employed the same Candidate Image Descriptions as in Figure 18 to create a new prompt and generate a response.
>     - The resulting response was inferior in quality, failing to meet the desired formatting criteria, such as assigning image IDs, specifying the number of images per dialogue turn, and incorporating new images in subsequent turns. Furthermore, the response lacked the diversity that its dialogues typically ask for more detailed descriptions of images but not specifying particular aspects. In conclusion, dialogue demonstrations are crucial not only for enhancing data quality by providing formatting guidelines but also for increasing diversity by conditioning different demonstrations.
>
> 3. **Baseline with Strong Caption Model and Instruction-Tuned LM**
>    We appreciate your suggestion to consider a baseline using a strong caption model combined with an instruction-tuned language model. We analyze the advantages of multimodal models over this baseline lie in that (1) some images are challenging to describe solely through language, as visual elements can be more expressive than words, and (2) multimodal models can interpret and discuss visual elements in specific contexts. In light of your suggestion, we plan to experiment with this in future iterations to provide a more comprehensive comparison.
>
> 4. **Capability with More than Two Images**:
>    - SparklesChat is capable of handling dialogues involving multiple images, not limited to two. **We include examples with four or five images in Appendix F of the revised manuscript to demonstrate this expanded capability.**
>     - To analyze SparklesChat's ability to generalize in scenarios involving a larger number of images, we consolidated several questions from a dialogue into a single query. This was done to generate responses for queries involving four or five images, as shown in **Figure 9 and Figure 10** in the updated manuscript, where the model effectively distinguishes between the images and adheres to complex queries. In the case of Figure 10, where three questions involving five images are concatenated into one query, the model tends to ignore the final question and only responds to the first two. We believe this limitation arises from the absence of similar patterns in the training data. A potential solution could involve incorporating multiple turns into each training dialogue to enhance the model's ability to handle such complex scenarios.

---

### Official Review · Reviewer_9M6X · 2023-10-31

**Soundness:** 2 fair
**Presentation:** 2 fair
**Contribution:** 2 fair
**Rating:** 5
**Confidence:** 4

**Summary:**

This work introduces SparklesChat, a multimodal instruction-tuned model designed to effectively engage in dialogues that encompass multiple images. Additionally, the constructed multi-image dialogue dataset and an evaluation benchmark are introduced.

**Strengths:**

This work focuses on a new scenario that is not well-explored by current large multimodal models, i.e. multi-image multimodal dialogue.

This work propose new training data, evaluation benchmark, and model for this scenario, which exhibit better performance than MiniGPT-4.

**Weaknesses:**

**1. The data construction process seems too trivial and not sound.**

In the data construction process to generate visual dialogue with multiple images, you provide multiple image-text pairs and ask GPT-4 to link them together, which I think is the simplest way to construct multi-image dialogues.

Besides, this simple approach fails to yield effective samples. In Figure 3, the response from GPT-4 seems too naive, *i.e.*, in image #1, we see ..., in image #2, we witness..... This is just a concatenation of descriptions of two images.

**2. Insufficient experiments.**

I think current experiments cannot form a strong foundation to support the effectiveness of your model and training data.

* Baselines. You compare your method only with MiniGPT-4, which in my understanding is an embarassingly weak and simple model & dataset. More comparisons are definitely needed.

* Evaluation benchmarks. You use three benchmarks for evaluation, BISON, NLVR2, and your own evaluation data. Among them, BISON and NLVR2 are not commonly used benchmarks now. Besides, on your own evaluation data, you claim your performance apporach GPT-4. However, your self-constructed training data could share similar distribution to you eval data. To this end, I think this claim cannot well establish.

**Questions:**

More solid experiments could be helpful.

---

> ### Author Response · Authors · 2023-11-23
> **Response to Reviewer 9M6X**
>
> Thank you for your valuable feedback! We appreciate your recognition of the novelty of our scenario and our contributions to the field in introducing new training data and benchmarks. Below, we address your concerns regarding the data construction process and our experimental setup.
>
> ---
>
> 1. **Data Construction Process:**
>    - Our approach to generating visual dialogue with multiple images extends beyond simple image-text pair linking or descriptions. We instruct GPT-4 to simulate realistic user-assistant interactions as introduced in Section 4.1, and not to directly describe images with the prompt *"Please make sure not to reveal the content of the images or describe the images in the user messages"* as shown in Table 8. Thus, the generated dialogues are not mere concatenations of descriptions but are contextually rich and coherent conversations.
>    - Sometimes, a detailed image description is part of the image understanding and reasoning process. The case in Figure 3 is picked to show a response that clearly distinguishes between multiple images. Meanwhile, Figure 19 (original Figure 17) presents a more complex example of generating a promotional text inspired by several images but not describing them naively.
>    - The statistics in Figure 4 further illustrate the diversity of SparklesDialogue, showcasing its range from generating text materials to seeking advice or discussing relationships between images. This involves reasoning about the content and context of the images, connecting different visual elements, and generating responses that reflect a deep understanding of the images about the user's queries.
>
> 2. **Expanded Experiments and Benchmarks:**
>
>     Thanks for your constructive suggestion!
>
>     **Expanded Baselines.** SparklesChat is built upon the MiniGPT-4 and adapted to accommodate multiple images due to its relevance and prominence in the field. We **extended our approach to more advanced models such as LLaVA-v1.5** to offer a broader view of our approach's effectiveness. LLaVA-v1.5 has improved its image input resolution from 224 pixels to 336 pixels. It is trained on a diverse dataset comprising about 665K instruction-following data. Additionally, LLaVA-v1.5* is fine-tuned on SparklesDialogue (about 6.5K) using the low-resource technique LoRA to conserve memory and time. The results are shown below and have **updated in the revised manuscript in Section 5.1 and Table 2.**
>
>     | Model        | Instruction Data| BISON | NLVR2 | SparklesEval |
>     |--------------|-----------------|-------|-------|-------|
>     | GPT-4        | -               | -     | -     | 9.26  |
>     | MiniGPT-4    | description     | 46.0% | 51.3% | 3.91  |
>     | MiniGPT-4*   | description     | 51.3% | 46.7% | 3.50  |
>     | LLaVA-v1.5   | Mixture (665K ) | 52.7% | 53.3% | 2.75  |
>     | LLaVA-v1.5*  | **+SparklesDialogue**| 65.3% | 56.7% | 7.93  |
>     | SparklesChat | description     | 52.0% | 48.0% | 3.06  |
>     | SparklesChat | reasoning       | 52.7% | 54.0% | 6.71  |
>     | SparklesChat | SparklesDialogue| 56.7% | 58.0% | 8.56  |
>
>     According to the results, despite LLaVA-v1.5's advantages of higher image resolution (336 vs. 224 pixels) and a larger training set (665K vs. 6.5K), SparklesChat significantly outperforms LLaVA-v1.5 in three evaluation sets involving multiple images. While LLaVA-v1.5 outperforms MiniGPT-4 on BISON and NLVR2, it shows weaker results on SparklesEval. This may be due to LLaVA-v1.5's training data primarily focusing on closed-set multimodal tasks such as VQA, TextCaps, and RefCOCO, while lacking in open-ended dialogue training. After fine-tuning with our SparklesDialogue, LLaVA-v1.5* not only significantly improved in open-ended dialogue tasks but enhanced traditional multimodal tasks. These results validate the adaptability of our method in unlocking chats across multiple images for multimodal instruction-following models with minimal additional training cost.
>
>     **Evaluation Benchmarks.** Our SparklesEval benchmark is designed to evaluate open-ended dialogues involving multiple images and turns, using LLMs as judges. Additionally, NLVR and BISON were specifically chosen for their use of natural images and the feasibility of automatic evaluation with unambiguous answers, such as TRUE/FALSE or single-choice problems. In response to your suggestion, we are expanding our experimental framework to include broader evaluation benchmarks, incorporating images from diverse domains (e.g., documents, fashion, cartoons) and a wider range of evaluation metrics beyond accuracy measures. Additionally, we plan to diversify SparklesEval with different distributions of images and text to reduce potential biases.

---

### Author Response · Authors · 2023-11-23
**Updated Manuscript and Response to All Reviewers**

**Thank you to all the reviewers for their insightful feedback and constructive comments!**

We are grateful to **most reviewers** (`R#1`, `R#2`, `R#4`) for their high recognition of one of our first explorations of a new scenario that addresses a key limitation of current LLMs, namely multi-image multimodal dialogue. We are pleased to hear that **all reviewers** acknowledge the introduction of our new dataset, SparklesDialogue, and the SparklesEval benchmark. Additionally, we sincerely thank Reviewer `R#2` for recognizing the novelty of our fine-grained integration of images and text, Reviewers `R#2` and `R#3` for their positive comments on our experiments' improved performance over MiniGPT-4, and Reviewer `R#4` for finding the Appendix and supplementary materials helpful and thorough.

We have carefully considered each of the concerns and suggestions raised by the reviewers and have undertaken revisions to our manuscript in response. The main revisions, **highlighted in blue** for clarity, include:
- Updated Section 5.1 and Table 2 to include experiments comparing our model with and extending our approach to another model, LLaVA-v1.5.
- Added examples of dialogues with four or five images in Figures 9 and 10 of Appendix F, showcasing our model's expanded capabilities.
- Added examples and analysis in Appendix K to demonstrate the contribution of dialogue demonstrations to data quality and diversity.
- Corrected a typographical error and relocated some detailed model training information to an appendix to save space.

Once again, we thank all the reviewers for their time and invaluable contributions and hope it is not too late to address your concerns effectively. We remain committed to engaging constructively with your valuable feedback.

*Please note that for simplicity, in our response, R#1, R#2, R#3, and R#4 refer to Reviewer 9M6X, Reviewer viMW, Reviewer eU8x, and Reviewer qmnH, respectively.*

---

### Meta-Review · Area_Chair_iq1b · 2023-12-13

**Metareview:**

This paper introduces a multimodal instruction following model for open-ended dialogues across multiple images. After rebuttal, it received scores of 5566. Generally, this is a borderline case.

On one hand, the topic studied in this paper can be important. It is natural to extend sing-image understanding to multi-image understanding. The paper provides a training set, evaluation set, and a trained model. On the other hand, reviewers commented that (1) the data construction process seems not sound and the conversations in the dataset seem to have a typical pattern with little diversity. Higher-quality datasets are needed, as the model is just MiniGPT-4. (2) The experimental section is a little bit weak, more results on more benchmarks are desired.

Overall, the AC decided to recommend rejection by the end.

**Justification For Why Not Higher Score:**

The model contains no novelty, the data construction process seems not very sound, and experimental results are not sufficient enough.

**Justification For Why Not Lower Score:**

N/A

---

### Decision · Program_Chairs · 2024-01-16

Reject